# Graph is All You Need? Lightweight Data-agnostic Neural Architecture Search without Training

## Abstract

Neural architecture search (NAS) enables the automatic design of neural network models. However, training the candidates generated by the search algorithm for performance evaluation incurs considerable computational overhead. Our method, dubbed *NASGraph*, remarkably reduces the computational costs by converting these neural architectures to graphs, and properties of the converted graphs are used as the proxy scores in lieu of validation accuracy. Our training-free NAS method is data-agnostic and light-weight. It can find the best architecture among 200 randomly sampled architectures from NAS-Bench201 in **217 CPU seconds**. We are able to achieve state-of-the-art performance on 7 out of 9 datasets in NASBench-101, NASBench-201, and NDS search spaces. We also demonstrate that *NASGraph* generalizes to more challenging tasks on Micro TransNAS-Bench-101.

## 1 Introduction

Neural architecture search (NAS) aims to automate the process of discovering state-of-the-art (SOTA) deep learning models. The objective of NAS is to find an optimal neural architecture $a^* = \arg\min_{a \in \mathcal{A}} f(a)$, where $f(a)$ denotes the performance (e.g., a task-specific loss function) of the neural architecture $a$ trained for a fixed number of epochs using a dataset, and $\mathcal{A}$ is the search space. Previous NAS techniques achieve competitive performance in various applications such as image classification Zoph et al. (2018); Real et al. (2019); Tan & Le (2019), object detection Chen et al. (2019); Ghiasi et al. (2019) and semantic segmentation Liu et al. (2019). The pioneering work Zoph & Le (2016) based on reinforcement learning is resource intensive, requiring more than 20k GPU days. In particular, training the generated neural architecture candidates to evaluate their performance is the most computationally expensive step. To accelerate the search process, various proposals have been made, including weight sharing Pham et al. (2018a); Chu et al. (2021), progressive complexity search stage Liu et al. (2018a), gradient descent in the differentiable search space Liu et al. (2018c); Liang et al. (2019); Xu et al. (2019), predictor-based NAS techniques White et al. (2021); Shi et al. (2020), etc. The recent emergence of training-free NAS Mellor et al. (2021); Abdelfattah et al. (2021); Lin et al. (2021); Chen et al. (2021) pushes the boundary of efficient NAS techniques further and greatly eases the computational burden. Training-Free NAS computes a proxy metric in place of accuracy to rank the candidate architectures. The proxy metric is obtained by a single forward/backward propagation using a training dataset.

In this paper, we take a novel perspective on mapping neural networks to graphs: we treat inputs to neural components as graph nodes and use relationship between inputs to neighboring graph components to determine the connectivity between pairs of graph nodes. In this way, we are able to convert a neural architecture to a DAG $G(V, E)$ where node set $V = \{v_1, v_2, \ldots, v_n\}$ and edge set $E = \{e_{ij}\} \, \forall i, j$ s.t. there exists an edge from $v_i$ to $v_j$. After establishing the mapping, we extract the associated graph measures as NAS metrics to rank neural architectures. We note that the entire process is training-free and data-agnostic (i.e. do not require any training data).

We summarize our **main contributions** as follows:

- We propose *NASGraph*, a training-free and data-agnostic method for NAS. *NASGraph* maps the neural architecture space to the graph space, which offers new insights into graph-oriented analysis for neural networks.

- Using the extracted graph measures for NAS, *NASGraph* achieves competitive performance on NAS-Bench-101, NAS-Bench-201, Micro TransNAS-Bench-101 and NDS benchmarks, when compared to existing training-free NAS methods. The analysis of bias towards operations in the benchmarks indicates that *NASGraph* has the lowest bias compared to those methods.

- In comparison to existing training-free NAS techniques, we show that the computation of *NASGraph* is lightweight (only requires CPU).

## 2 RELATED WORK

**One-shot NAS.** One-shot NAS constructs a supernet subsuming all candidate architectures in the search space. In other words, subnetworks of the supernet are candidate architectures. The supernet method is faster than the conventional NAS methods because it enables weight sharing among all the candidate architectures in the search space. The first one-shot NAS algorithm Pham et al. (2018a) considers the operations at each edge to be a discrete decision. DARTS Liu et al. (2018c) devises a continuous relaxation by formulating the value of the operation applied to an edge as a weighted sum of all the operations. Several works, e.g. Cai et al. (2019), Xie et al. (2018), Xu et al. (2019), Wang et al. (2021), Chu et al. (2020), Zela et al. (2020) were developed to improve some shortcomings of DARTS such as memory consumption, DARTS collapse and increase in the validation loss after the discretization step. Yu et al. (2020) demonstrated the discrepancy between the ranking of architectures sampled from a trained supernet and that of the same architectures when trained from scratch. To alleviate co-adaptation among the operations, works such as Guo et al. (2020), Bender et al. (2018), Chu et al. (2021) recommend uniformly sampling the candidate architectures while others recommended pruning the search space during the search Noy et al. (2020). To accelerate the search process, EcoNAS Zhou et al. (2020) proposes a hierarchical proxy strategy that uses a faster proxy for less promising candidates and a better but slower proxy for more promising candidates.

**Training-Free NAS.** Training-Free NAS uses models with randomly initialized weights to obtain the saliency metrics that rank these models. Since there is no need for training models, this routine is considerably faster even compared to one-shot NAS. NASWOT Mellor et al. (2021) applies the theory on the linear regions in deep networks Hanin & Rolnick (2019) to achieve NAS without training. The saliency metrics for the pruning-at-initialization work in network pruning are also found to be effective in zero-cost NAS Abdelfattah et al. (2021). In addition, TENAS Chen et al. (2021) uses metrics from the condition number of neural tangent kernel and the number of linear regions. A comparison of saliency metrics Krishnakumar et al. (2022) finds that these metrics might contain complementary information and hence a combination of metrics can be helpful in NAS.

## 3 NASGRAPH: A GRAPH-BASED FRAMEWORK FOR DATA-AGNOSTIC AND TRAINING-FREE NAS

Figure 1 shows an overview of our proposed *NASGraph* framework. A neural architecture is uniquely mapped to a DAG, i.e. $a \mapsto G(V, E)$. After the conversion, graph measures are computed to rank the performance of the corresponding neural architectures. We also note that our notion of graph refers to the specific graph representation of a neural architecture $a$ via our proposed graph conversion techniques, instead of the original computation graph of $a$.

**Graph block.** The basic element in the *NASGraph* framework is the graph block. We use the notation $f^{(l)}(\mathbf{x}^{(l)}; \theta^{(l)})$ to represent the $l$-th graph block, where $\mathbf{x}^{(l)}$ is the input to the graph block and $\theta^{(l)}$ is the model parameter. We combine several neural components (e.g. Conv and ReLU) to a graph block such that the output of the graph block $y^{(l)} = f^{(l)}(\mathbf{x}^{(l)}; \theta^{(l)})$ is non-negative. Table 1 lists different graph blocks used in this paper. We note that our graph block can be extended to cover different combinations of neural components as long as the block output is non-negative.

**Conversion method.** Inspired by the iterative re-evaluation method in the network pruning Verdenius et al. (2020); Tanaka et al. (2020), we apply conversion to each graph block independently. Similarly, we also use all-ones matrix as the input to the graph block $\mathbf{x}^{(l)} = \mathbb{1}^{C^{(l-1)} \times H^{(l-1)} \times W^{(l-1)}}$ in the forward propagation process, where $C^{(l-1)}$ is the number of channels, $H^{(l-1)}$ is the image height,

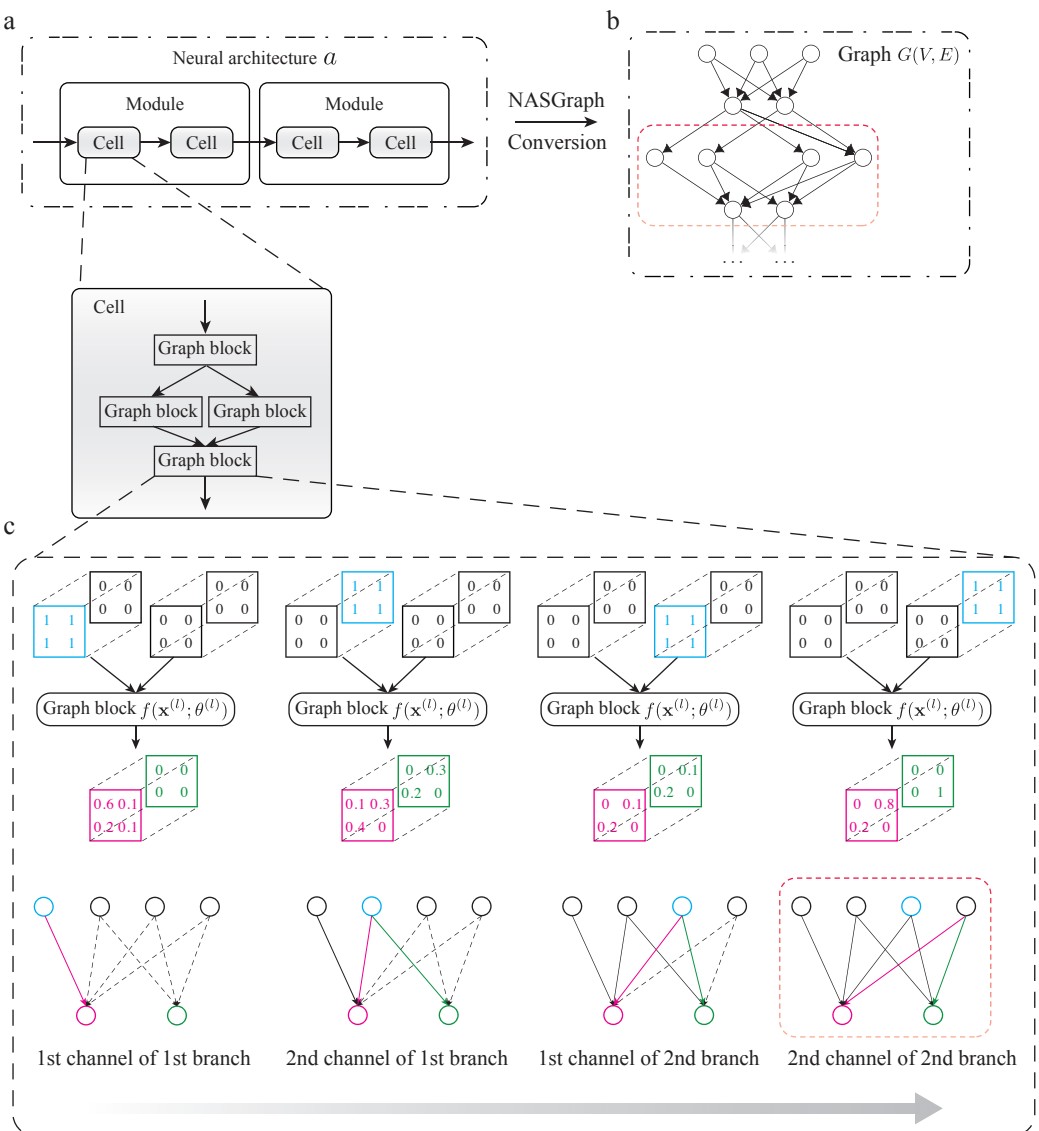

Figure 1: An overview of the *NASGraph* framework: the connectivity of graph nodes is determined by the forward propagation of the corresponding graph blocks. In the toy example shown in the bottom of the figure, if the output from the forward propagation is all-zeros matrix $\mathbb{O}$, there is no connection. Otherwise, the connection is built between a pair of graph nodes. The orange rectangles in (b) and (c) mark how a subgraph generated by a single forward propagation constitutes a part of the whole graph.

and $W^{(l-1)}$ is the image width for the input to $l$-th graph block. This helps us get an unbiased estimate of the contribution of the input to the output. Further, to determine the contribution of the $c$-th channel of the input on the channels of the output for the $l$-th graph block, we apply a mask $\mathcal{M}_c^{(l)}$ to the input so that only the $c$-th channel $(\mathbf{x}_{d_1 d_2 d_3}^{(l)})_{d_1=c}$ is an all-ones matrix $\mathbb{1}^{H^{(l-1)} \times W^{(l-1)}}$ and other channels are zero matrices $\mathbb{O}^{H^{(l-1)} \times W^{(l-1)}}$. A toy example is shown in the bottom of Figure 1. We evaluate the contribution of the $c$-th channel $(\mathbf{x}_{d_1 d_2 d_3}^{(l)})_{d_1=c}$ to the output $\mathbf{y}^{(l)}$ by performing a forward propagation as described by:

$$\mathbf{y}_c^{(l)} = f^{(l)}(\mathcal{M}_c^{(l)} \odot \mathbf{x}^{(l)}; \theta^{(l)}) \tag{1}$$

Table 1: Diverse graph blocks in the framework of *NASGraph*. The sign of the output of the graph block is determined by passing an artificial non-negative input.

| Graph blocks | Number of components | Sign of output | Parametric* |
|---|---|---|---|
| Conv-BN-ReLU | 3 | Non-negative | True |
| Conv-Conv-BN-ReLU | 4 | Non-negative | True |
| MaxPool | 1 | Non-negative | False |
| AvgPool | 1 | Non-negative | False |
| Identity | 1 | Non-negative | False |
| Zero | 1 | Zero | False |

∗ A graph block is parametric if it contains model parameters that will be updated in the backward propagation process when training a neural network model. Note that there is no backpropagation in *NASGraph* framework.

where $f^{(l)}(\cdot)$ is the $l$-th graph block, $\odot$ is the Hadamard product, and $\theta^{(l)}$ is the parameters of the $l$-th graph block. The score $\omega_{i^{(l-1)}j^{(l)}}$ for the edge $e_{ij}$ between node $i^{(l-1)}$ and $j^{(l)}$ is determined by:

$$\omega_{i^{(l-1)}j^{(l)}} = \sum_{d_2=1}^{H^{(l)}} \sum_{d_3=1}^{W^{(l)}} ((y_i^{(l)})_{d_1 d_2 d_3})_{d_1=j} \tag{2}$$

If $\omega_{i^{(l-1)}j^{(l)}}$ is larger than 0, we build an edge between node $i^{(l-1)}$ and node $j^{(l)}$ that indicates the relationship between $i$-th channel of the input $\mathbf{x}^{(l)}$ and $j$-th channel of the output $\mathbf{y}^{(l)}$. We use a virtual input graph block of identity operation to take the input to the neural architecture into consideration. After looping though all graph blocks, we can uniquely construct a graph.

**Analysis of the proposed method.** We consider a neural component consisting of convolution and ReLU activation function. We use $\mathcal{F}$ to represent the neural component, $\mathcal{F}(\mathbf{x}) = h \circ f(\mathbf{x})$. The function $f(\cdot)$ is the convolution operation and $h(\cdot)$ is the ReLU activation function.

**Theorem 1.** *Let $\mathbf{x}$ be the input to $\mathcal{F}$. $\mathcal{F}$ is converted to a graph $G(V, E)$ using NASGraph framework: the input to $\mathcal{F}$ has $\mathbb{1} \in \mathbb{R}^{H^{(0)} \times W^{(0)}}$ for the channel $i$ and $\mathbb{0} \in \mathbb{R}^{H^{(0)} \times W^{(0)}}$ for rest of channels. There is no edge between node $i$ and node $j$ if and only if the output for the channel $j$ is all-zeros matrix $\mathbb{0} \in \mathbb{R}^{H^{(1)} \times W^{(1)}}$.*

We provide the proof in Appendix A.2. Recall that we use $\mathbb{1}$ as input to $\mathcal{F}$ to make an unbiased estimate on the contribution of the input to the output. Hence, an output of $\mathbb{0}$ means there exists a contribution from the input to the output, which is inferior to the performance of neural architecture. Theorem 1 indicates that the connectivity between graph nodes is associated with the neural architecture performance. Hence, we can use graph measures that are sensitive to node connectivity as the metrics to rank neural architectures.

**Improving the search efficiency of NASGraph.** To reduce computational overhead, NAS typically uses a training-reduced proxy to obtain the performance of neural architectures. A systematic study is reported in EcoNAS Zhou et al. (2020) where four reducing factors are analyzed: (1) number of epochs, (2) resolution of input images, (3) number of training samples, (4) number of channels for Convolution Neural Networks (CNNs). Following their convention, to accelerate *NASGraph*, we also consider the surrogate models, i.e. models with computationally reduced settings. We dub the surrogate model *NASGraph*(**h, c, m**), where $h$ is the number of channels, $c$ is the number of search cells in a module, and $m$ is the number of modules. The ablation study on the effect of using a surrogate model is discussed in Appendix A.6.

After converting the neural architectures to graphs, we use the average degree as the graph measure to rank neural architectures. In addition to the average degree, we also examine the performance of other graph measures. The result is included in the Appendix A.8.

## 4 PERFORMANCE EVALUATION

### 4.1 EXPERIMENT SETUP

**NAS benchmarks.** To examine the effectiveness of our proposed *NASGraph*, we use neural architectures on NAS-Bench-101 Ying et al. (2019), NAS-Bench-201 Dong & Yang (2020), TransNAS-Bench-

101 Duan et al. (2021) and Network Design Space (NDS) Radosavovic et al. (2019). NAS-Bench-101 is the first NAS benchmark consisting of 423,624 neural architectures trained on the CIFAR-10 dataset Krizhevsky et al. (2009). The training statistics are reported at 108th epoch. NAS-Bench-201 is built for prototyping NAS algorithms. It contains 15,625 neural architectures trained on CIFAR-10, CIFAR-100 Krizhevsky et al. (2009) and ImageNet-16-120 Chrabaszcz et al. (2017) datasets. The training statistics are reported at 200th epoch for these three datasets. In addition to standard NAS benchmarks, we also examine the performance of *NASGraph* on TransNAS-Bench-101, specifically the micro search space. The micro (cell-level) TransNAS-Bench-101 has 4,096 architectures trained on different tasks using the Taskonomy dataset Zamir et al. (2018). The NDS benchmark includes AmoebaNet Real et al. (2019), DARTS Liu et al. (2018c), ENAS Pham et al. (2018b), NASNet Zoph & Le (2016) and PNAS search spaces Liu et al. (2018b).

We convert neural architectures with Gaussian initialization to graphs $G(V, E)$. The conversion is repeated 8 times with different random initializations. We find the difference in the rankings among 8 initializations is considerably marginal as shown in Appendix A.7.

We use AMD EPYC 7232P CPU in the computation of *NASGraph*. To compute the performance of baselines requiring GPUs, a single NVIDIA A40 GPU is used. We reduce the number of channels to be 16 and the number of cells to be 1, i.e. *NASGraph*(16, 1, 3) is used as the surrogate model.

**Baselines.** We use metrics in training-free NAS as our baselines. relu_logdet Mellor et al. (2021) applies the theory on the number of linear regions to represent the model expressivity. jacob_cov Mellor et al. (2021) is based on the correlation of Jacobians with inputs. grad_norm Abdelfattah et al. (2021) sums the Euclidean norm of the gradients. snip Lee et al. (2018) is related to the connection sensitivity of neural network model. grasp Wang et al. (2020) is based on the assumption that gradient flow is preserved in the efficient training. fisher Theis et al. (2018) estimates fisher information of model parameters, synflow Tanaka et al. (2020) preserves the total flow of synaptic strength.

## 4.2 Scoring neural architectures using average degree

The distribution of the average degree of graphs vs the performance of neural architectures is plotted in Figure 2 (a)-(l). Colorbar indicates the number of architectures in logarithmic base 10. We compute both Spearman's ranking correlation $\rho$ and Kendall's Tau ranking correlation $\tau$ between them. In Table 2 we compare the performance of average degree against all the baselines. Our method can rank architectures effectively and outperforms the baselines on most of the datasets on NAS-Bench-101 and NAS-Bench-201.

We also evaluate our technique on the NDS benchmark. The result is summarized in Table 4. Similar to NASWOT Mellor et al. (2021), we use 1000 randomly sampled architectures. Our method is able to ourperform all the other baselines on 4 search spaces and comes second on the DARTS search space.

To explore the generality of the proposed *NASGraph* framework, we examine the performance of the graph measure on micro TransNAS-Bench-101. The comparison is shown in Table 3. As reported in Krishnakumar et al. (2022), there is a pronounced variation in the ranks of training-free proxies. For the *class_object* downstream task, the average degree gives the best performance. For other downstream tasks, our method also exhibits a competitive performance.

## 4.3 Combining NASGraph metrics with data-dependent NAS metrics improves prediction

In addition to using single metric, we examine the performance of combining graph measures with existing metrics. Specifically, two metrics are combined by the summation of rankings of neural architectures by two metrics. We use a combination of avg_deg and jacob_cov, i.e. rank(avg_deg) + rank(jacob_cov). For the case of tied ranking, the average of the ranking is used. Similar to density and average degree, the combined metrics manifest a positive correlation with test accuracy. By combining our metrics with jacob_cov, the $\rho$ boosts to 0.85 on CIFAR-10 and CIFAR-100 and the $\tau$ reaches 0.66 on CIFAR-10, 0.67 on CIFAR-100. On ImageNet-16-120, the $\rho$ increases to 0.82 and $\tau$ is 0.64. The comparison with existing combined metrics is shown in Table 5. Our combined metrics can outperform all existing methods reported in Chen et al.

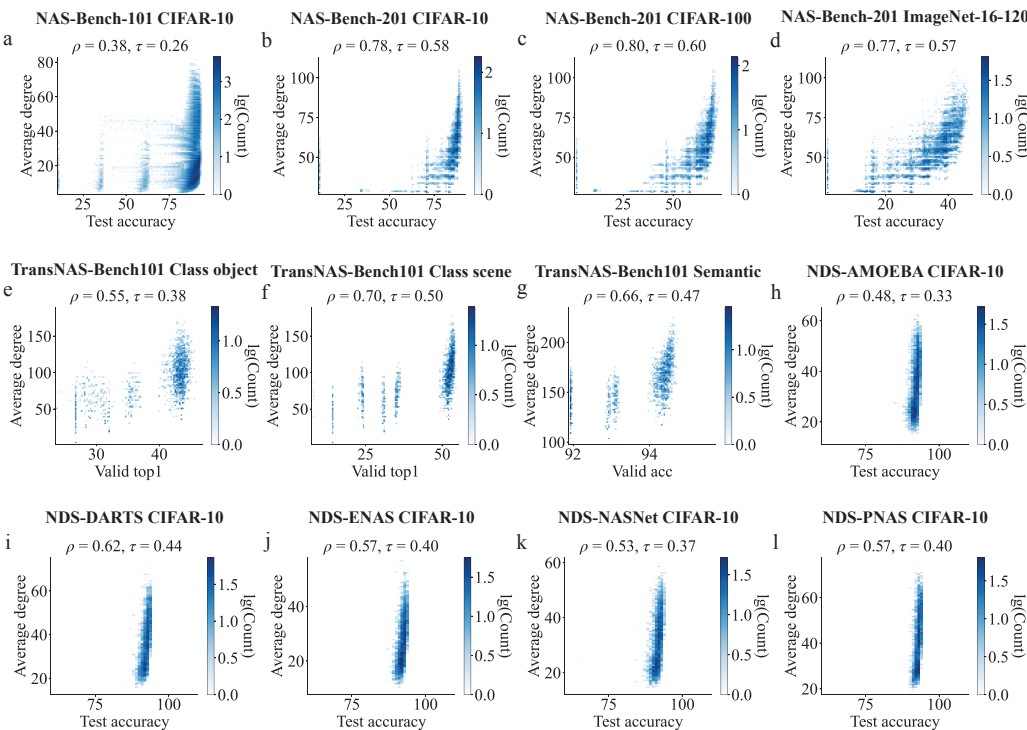

Figure 2: The ranking correlation between the performance of neural architecture $a$ and the graph measures of the corresponding graph $G(V, E)$. In the *NASGraph* framework, each neural architecture is uniquely mapped to a graph, i.e. $a \mapsto G(V, E)$.

Table 2: Comparison of the ranking correlation between *NASGraph* and training-free NAS methods using single metric. Correlations are calculated between the metrics and test accuracies on different NAS benchmarks and datasets.

| Method | Metric | NAS-Bench-101 CIFAR-10 | | NAS-Bench-201 CIFAR-10 | | CIFAR-100 | | ImageNet-16-120 | |
|---|---|---|---|---|---|---|---|---|---|
| | | $\rho$ | $\tau$ | $\rho$ | $\tau$ | $\rho$ | $\tau$ | $\rho$ | $\tau$ |
| NASWOT | relu_logdet | - | **0.31** | 0.76 | 0.57 | 0.79 | **0.61** | 0.71 | 0.55 |
| TENAS[‡] | NTK | - | - | - | - | - | -0.42 | - | - |
| | NLR | - | - | - | - | - | -0.50 | - | - |
| Zero-Cost NAS | grad_norm | 0.20 | - | 0.58 | 0.42 | 0.64 | 0.47 | 0.58 | 0.43 |
| | snip | 0.16 | - | 0.58 | 0.43 | 0.63 | 0.47 | 0.58 | 0.43 |
| | grasp | 0.45 | - | 0.48 | 0.33 | 0.54 | 0.38 | 0.56 | 0.40 |
| | fisher | 0.26 | - | 0.36 | 0.26 | 0.39 | 0.28 | 0.33 | 0.25 |
| | synflow | 0.37 | - | 0.74 | 0.54 | 0.76 | 0.57 | 0.75 | 0.56 |
| | jacob_cov | 0.38 | - | 0.73 | 0.55 | 0.71 | 0.55 | 0.71 | 0.54 |
| Ours | avg_deg | **0.38** | 0.26 | **0.78** | **0.58** | **0.80** | 0.60 | **0.77** | **0.57** |

‡ TENAS examines two metrics. One metric is the condition number of the neural tangent kernel (NTK). Another metric is the number of linear regions (NLR). As the code to compute the correlations has not been released, we cannot compute correlations on other datasets.

(2021); Abdelfattah et al. (2021), which demonstrates the effectiveness of complementary nature of our proposed graph measures for NAS. The detailed plots of combined ranks against test accuracy are shown in Appendix A.9.

## 4.4 TRAINING-FREE ARCHITECTURE SEARCH USING NASGRAPH

We evaluate the performance of our metric when used as an alternative to validation accuracy during random search. $N = 100$ and $N = 200$ architectures are randomly sampled from NAS-Bench-201

Table 3: Comparison of Spearman's ranking correlations $\rho$ between validation accuracies and training-free NAS metrics on micro TransNAS-Bench-101. The baseline performance is extracted from Krishnakumar et al. (2022).

| Metric | Micro TransNAS-Bench-101 | | | | |
| | class_object | class_scene | room_layout | segment_semantic | Rank mean |
|---|---|---|---|---|---|
| plain | 0.34 | 0.24 | 0.36 | -0.02 | 8.50 |
| grasp | -0.22 | -0.27 | -0.29 | 0.00 | 9.75 |
| fisher | 0.44 | 0.66 | 0.30 | 0.12 | 6.50 |
| epe_nas | 0.39 | 0.51 | 0.40 | 0.00 | 7.25 |
| grad_norm | 0.39 | 0.65 | 0.25 | 0.60 | 6.75 |
| snip | 0.45 | 0.70 | 0.32 | 0.68 | 4.25 |
| synflow | 0.48 | 0.72 | 0.30 | 0.00 | 5.75 |
| l2_norm | 0.32 | 0.53 | 0.18 | 0.48 | 8.75 |
| params | 0.45 | 0.64 | 0.30 | 0.68 | 5.25 |
| zen | 0.54 | 0.72 | 0.38 | 0.67 | 2.50 |
| jacob_cov | 0.51 | 0.75 | 0.40 | 0.80 | 1.50 |
| flops | 0.46 | 0.65 | 0.30 | 0.69 | 4.50 |
| naswot | 0.39 | 0.60 | 0.25 | 0.53 | 7.50 |
| avg_deg (Ours) | 0.55 | 0.70 | 0.37 | 0.66 | 3.00 |

Table 4: Comparison of Kendall's Tau ranking correlations $\tau$ between test accuracies and training-free NAS metrics on the NDS benchmark.

| Metric | AMOEBA | DARTS | ENAS | NASNet | PNAS |
|---|---|---|---|---|---|
| grad_norm | -0.12 | 0.22 | 0.06 | -0.05 | 0.15 |
| snip | -0.09 | 0.26 | 0.10 | -0.02 | 0.18 |
| grasp | 0.02 | -0.04 | 0.03 | 0.18 | -0.01 |
| fisher | -0.12 | 0.19 | 0.04 | -0.07 | 0.14 |
| jacov_cov | 0.22 | 0.19 | 0.11 | 0.05 | 0.10 |
| synflow | -0.06 | 0.30 | 0.14 | 0.04 | 0.21 |
| naswot | 0.22 | **0.47** | 0.37 | 0.30 | 0.38 |
| avg_deg (Ours) | **0.32** | 0.45 | **0.41** | **0.37** | **0.40** |

and the training-free metrics are used to perform the search. We repeat the search process 100 times, and the mean and the standard deviation are reported in Table 6. Ground truth (GT) indicates the highest validation accuracy and the highest test accuracy for the validation and the test columns respectively. It is essential to highlight the fact that all the metrics in baselines are computed based on single A40 GPU (in GPU second) while *NASGraph* score is calculated on single CPU (in CPU second). *NASGraph* using the surrogate model NASGraph(16, 1, 3) can outperform the other training-free metrics on CIFAR-10 and CIFAR-100 datasets with a higher mean value and a lower standard deviation. At a small cost of performance, the surrogate model NASGraph(1, 1, 3) can have a significant improvement in the computation efficiency.

## 4.5 ANALYSIS OF OPERATION PREFERENCE IN NAS METHODS

To study the bias in architecture preference in NAS, we count the frequency (histogram) of different operations appearing in the top 10% architectures (ranked by different metrics) on NAS-Bench-201. The visualization of the frequency distribution of operations and the details on frequency counting are given in Appendix A.3. The overall result is shown in Table 7. The groundtruth

Table 5: Comparison of the ranking correlations using multiple metrics with training-free NAS methods on NAS-Bench-201. Correlations between the combined metric and NAS Benchmark performance are reported. TENAS Chen et al. (2021) combines rankings by NTK and NLR. Zero-Cost NAS Abdelfattah et al. (2021) takes a majority vote among the three metrics: synflow, jacob_cov and snip. Our method combines the rankings of avg_deg and jacob_cov.

| Method | | CIFAR-10 | | CIFAR-100 | | ImageNet-16-120 | |
| | | $\rho$ | $\tau$ | $\rho$ | $\tau$ | $\rho$ | $\tau$ |
|---|---|---|---|---|---|---|---|
| TENAS | Rank combination | - | - | - | 0.64 | - | - |
| Zero-Cost NAS | Voting | 0.82 | - | 0.83 | - | 0.82 | - |
| Ours | Rank combination | **0.85** | 0.66 | **0.85** | **0.67** | **0.82** | 0.64 |

Table 6: Comparison of training-free NAS metrics in random search. The same subset of architectures is randomly chosen from NAS-Bench-201 for all metrics. GT reports the performance of the best architecture in that subset.

| Metric | Running time | CIFAR-10 | | CIFAR-100 | | ImageNet-16-120 | |
|---|---|---|---|---|---|---|---|
| | | validation | test | validation | test | validation | test |
| | | | | N = 100 | | | |
| relu_logdet | 52.72 GPU sec. | $89.51 \pm 0.96$ | $89.22 \pm 1.03$ | $69.48 \pm 1.44$ | $69.58 \pm 1.50$ | $42.92 \pm 2.41$ | $43.27 \pm 2.62$ |
| grad_norm | 364.68 GPU sec. | $88.28 \pm 1.42$ | $87.94 \pm 1.48$ | $65.96 \pm 3.11$ | $66.13 \pm 3.10$ | $34.97 \pm 6.82$ | $34.96 \pm 7.06$ |
| snip | 363.71 GPU sec. | $88.29 \pm 1.42$ | $87.95 \pm 1.48$ | $66.14 \pm 2.96$ | $66.32 \pm 2.97$ | $35.44 \pm 6.49$ | $35.44 \pm 6.72$ |
| grasp | 377.29 GPU sec. | $88.06 \pm 1.55$ | $87.74 \pm 1.58$ | $66.27 \pm 3.50$ | $66.38 \pm 3.55$ | $35.20 \pm 6.76$ | $35.19 \pm 6.93$ |
| fisher | 315.57 GPU sec. | $84.08 \pm 6.68$ | $83.70 \pm 6.67$ | $61.77 \pm 7.26$ | $61.89 \pm 7.44$ | $30.80 \pm 8.02$ | $30.49 \pm 8.33$ |
| synflow | 360.15 GPU sec. | $89.91 \pm 0.87$ | $89.67 \pm 0.88$ | $70.03 \pm 1.79$ | $70.17 \pm 1.79$ | $41.89 \pm 4.13$ | $42.23 \pm 4.24$ |
| jacob_cov | 360.48 GPU sec. | $88.68 \pm 1.56$ | $88.32 \pm 1.59$ | $67.45 \pm 2.91$ | $67.57 \pm 3.03$ | $40.64 \pm 3.54$ | $40.76 \pm 3.77$ |
| avg_deg (NASGraph(1, 1, 3)) | 7.78 CPU sec. | $89.74 \pm 0.77$ | $89.53 \pm 0.75$ | $69.90 \pm 1.38$ | $70.01 \pm 1.43$ | $42.00 \pm 2.80$ | $40.73 \pm 4.14$ |
| avg_deg (NASGraph(16, 1, 3)) | 106.18 CPU sec. | $89.95 \pm 0.49$ | $89.73 \pm 0.52$ | $70.17 \pm 1.06$ | $70.29 \pm 1.10$ | $42.72 \pm 2.33$ | $43.15 \pm 2.29$ |
| GT | - | $90.98 \pm 0.36$ | $90.77 \pm 0.31$ | $71.48 \pm 0.86$ | $71.69 \pm 0.81$ | $45.45 \pm 0.67$ | $45.74 \pm 0.65$ |
| | | | | N = 200 | | | |
| relu_logdet | 90.39 GPU sec. | $89.64 \pm 0.81$ | $89.33 \pm 0.84$ | $69.65 \pm 1.36$ | $69.87 \pm 1.33$ | $43.25 \pm 2.22$ | $43.62 \pm 2.37$ |
| grad_norm | 644.23 GPU sec. | $88.23 \pm 1.51$ | $87.87 \pm 1.53$ | $65.46 \pm 3.34$ | $65.67 \pm 3.42$ | $35.08 \pm 7.05$ | $35.00 \pm 7.26$ |
| snip | 712.58 GPU sec. | $88.23 \pm 1.51$ | $87.87 \pm 1.54$ | $65.68 \pm 3.16$ | $65.89 \pm 3.21$ | $35.08 \pm 7.05$ | $35.00 \pm 7.26$ |
| grasp | 692.74 GPU sec. | $88.31 \pm 1.35$ | $87.96 \pm 1.37$ | $65.97 \pm 3.21$ | $66.16 \pm 3.28$ | $34.83 \pm 6.63$ | $34.74 \pm 6.81$ |
| fisher | 622.92 GPU sec. | $85.55 \pm 4.91$ | $85.24 \pm 4.92$ | $61.69 \pm 5.62$ | $61.86 \pm 5.77$ | $29.39 \pm 6.38$ | $29.04 \pm 6.65$ |
| synflow | 742.74 GPU sec. | $89.87 \pm 0.85$ | $89.61 \pm 0.85$ | $69.93 \pm 1.84$ | $70.05 \pm 1.89$ | $41.54 \pm 3.76$ | $41.93 \pm 3.77$ |
| jacob_cov | 688.77 GPU sec. | $88.34 \pm 1.67$ | $88.00 \pm 1.71$ | $67.39 \pm 2.93$ | $67.55 \pm 3.05$ | $40.95 \pm 3.24$ | $41.04 \pm 3.41$ |
| avg_deg (NASGraph(1, 1, 3)) | 15.98 CPU sec. | $89.92 \pm 0.61$ | $89.69 \pm 0.62$ | $70.25 \pm 1.20$ | $70.42 \pm 1.21$ | $41.96 \pm 2.44$ | $42.48 \pm 2.39$ |
| avg_deg (NASGraph(16, 1, 3)) | 217.21 CPU sec. | $89.96 \pm 0.38$ | $89.73 \pm 0.43$ | $70.22 \pm 0.99$ | $70.45 \pm 0.98$ | $42.27 \pm 2.36$ | $42.76 \pm 2.36$ |
| GT | - | $91.14 \pm 0.25$ | $90.91 \pm 0.24$ | $71.84 \pm 0.76$ | $72.04 \pm 0.72$ | $45.72 \pm 0.54$ | $46.01 \pm 0.50$ |

(GT) ranks architecture using the test accuracy. A perfect metric is expected to have the same frequency distribution of operations as GT and thus achieve zero bias. We compute the accumulated difference in the frequency (also called bias) between a NAS metric and GT across all five operations in NAS-Bench-201. The avg_deg from *NASGraph* has the lowest bias when using a single metric. The combined metric (conm_rank) introduced in Section 4.3, which is rank(avg_deg) + rank(jacob_cov), can further reduce the bias. The low bias towards operations can also explain why our method can have a high rank correlation in NAS via *NASGraph*.

Table 7: Comparison of the accumulated frequency difference between training-free NAS methods and groundtruth (GT) on NAS-Bench-201 (the top 10% architectures ranked by a metric). GT ranks architectures ranked by test accuracies. Lower value means less bias (i.e. closer to GT).

| Dataset | relu_logdet | grad_norm | snip | grasp | fisher | synflow | jacob_cov | avg_deg | comb_rank |
|---|---|---|---|---|---|---|---|---|---|
| CIFAR-10 | 0.3 | 0.32 | 0.31 | 0.31 | 0.52 | 0.22 | 0.39 | 0.22 | 0.17 |
| CIFAR-100 | 0.27 | 0.3 | 0.29 | 0.28 | 0.52 | 0.18 | 0.42 | 0.14 | 0.17 |
| ImageNet-16-120 | 0.19 | 0.24 | 0.24 | 0.24 | 0.53 | 0.27 | 0.22 | 0.27 | 0.12 |
| Average bias | 0.25 | 0.29 | 0.28 | 0.28 | 0.52 | 0.22 | 0.34 | 0.21 | 0.15 |

All of the baselines except synflow rely on a few mini-batches of data. Thus, their score varies based on the input data being used to evaluate their proxy scores. In contrast, our metric is data-agnostic and is not sensitive to the input data. Furthermore, our technique is also lightweight: it can run on a CPU and the running time is faster than most of training-free proxies running on GPU.

# 5 DISCUSSION

In this paper, we proposed *NASGraph*, a novel graph-based method for NAS featuring lightweight (CPU-only) computation and is data-agnostic and training-free. Extensive experimental results verified the high correlation between the graph measure of *NASGraph* and performance of neural architectures in several benchmarks. Compared to existing NAS methods, *NASGraph* provides a more stable and accurate prediction of the architecture performance and can be used for efficient architecture search. We also show that our graph measures can be combined with existing data-dependent metrics to further improve NAS. We believe our findings provide a new perspective and useful tools for studying NAS through the lens of graph theory and analysis.

Our method addresses some limitations in current NAS methods (e.g. data dependency, GPU requirement, and operation preference) and attains new state-of-the-art NAS performance. The authors do not find any immediate ethical concerns or negative societal impacts from this study.

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
