# A APPENDIX

## A.1 CODE

Our code is available for review at: `https://anonymous.4open.science/r/nasgraph-61D7`.

## A.2 PROOF OF THEOREM 1

**Theorem 1.** *Let* $\mathbf{x}$ *be the input to* $\mathcal{F}$. $\mathcal{F}$ *is converted to a graph* $G(V, E)$ *using NASGraph framework: the input to* $\mathcal{F}$ *has* $\mathbb{1} \in \mathbb{R}^{H^{(0)} \times W^{(0)}}$ *for the channel* $i$ *and* $\mathbb{0} \in \mathbb{R}^{H^{(0)} \times W^{(0)}}$ *for rest of channels. There is no edge between node* $i$ *and node* $j$ *if and only if the output for the channel* $j$ *is all-zeros matrix* $\mathbb{0} \in \mathbb{R}^{H^{(1)} \times W^{(1)}}$.

**Proof:** $y_i^{(1)} = \mathcal{F}(\mathbf{x})$, and $(y_i^{(1)})_{d_1 d_2 d_3} \in \mathbb{R}$ is computed by:

$$(y_i^{(1)})_{d_1 d_2 d_3} = \text{ReLU}(\sum_{a=0}^{f_{k_1}^{(0)}} \sum_{b=0}^{f_{k_2}^{(0)}} \sum_{c=0}^{C^{(0)}} \mathbf{w}_{a,b,c}^{(0),d_3} \mathbf{x}_{a+(d_1-1)s_l,b+(d_2-1)s_l,c}^{(0)} + \mathbf{b}^{(0),d_3}) \tag{3}$$

where $f_{k_1}^{(0)}$ is the kernel height, $f_{k_2}^{(0)}$ is the kernel width, and $s_l$ is the convolution stride. Since $(y_i^{(1)})_{d_1 d_2 d_3} \geq 0$, according to Equation 2, $\omega_{i^{(l-1)}j^{(l)}} = 0$ implies $(y_i^{(1)})_{d_1 d_2 d_3} = 0$. The reverse implication is obvious: when $(y_i^{(1)})_{d_1 d_2 d_3} = 0$, $\forall d_1, d_2$ s.t. $0 \leq d_1 \leq H^{(1)}, 0 \leq d_2 \leq W^{(1)}$, then the summation $\omega_{i^{(l-1)}j^{(l)}} = 0$. This completes our proof.

## A.3 BIAS IN NAS METHODS

We rank architectures in NAS-Bench-201 using different metrics and compute the $\rho$ of architecture ranks for each pair of metrics. Figure 3 (a)-(c) show the pairwise ranking correlation between metrics. Both `avg_deg` and `synflow` are data-agnostic approach and the correlation between these two metrics is high. Besides, the correlations between these two metrics do no change across datasets. The correlation between `avg_deg` and `jacob_cov` is comparably small, but a combination of these two metrics can have the best ranking correlation between the metrics and the performance of neural architectures. Because of rounding number, `grad_norm` and `snip` have the $\rho$ of 1. But they do not give the exactly same rankings of neural architectures. When we check the architecture rankings precisely using these two metrics, we find $\rho = 0.9982$ on CIFAR-10, $\rho = 0.9984$ on CIFAR-100 and $\rho = 0.9988$ on ImageNet-16-120.

We extract top 10% neural architectures on NAS-Bench-201 and count the frequency of each operation (`avg_pool`, `none`, `nor_conv_1×1`, `nor_conv_3×3`, `skip_connect`) appearing in the selected subset. Figure 3 (e)-(f) show the frequency of different operations on different datasets. There is no pronounced difference in the distribution across different datasets for GT. This is reasonable when we examine the distribution of test accuracy in CIFAR-10, CIFAR-100 and ImageNet-16-120 datasets (Figure 2 (f)) where test accuracy is positively correlated among datasets. These facts indicate the same architecture generally have a similar performance on different datasets. Therefore, the data-agnostic NAS methods (`avg_deg` and `synflow`) can be effective to search neural architectures across different datasets.

Based on the distribution of operation preference, we find our metric `avg_deg` has a distribution similar to the GT. Similar to `synflow`, our metric has a relatively low preference for `skip_connect` while `jacob_cov` has a high preference for `skip_connect`. That might explain the reason why a combination of `avg_deg` and `jacob_cov` gives the highest correlation between test accuracy and combined metrics. Another reasoning is related to the definition of `jacob_cov`. The Jacobian for the $i$-th neuron in the output of the layer $L$ with parameter $\theta_\alpha$ valuated at a point $\mathbf{x}$ is defined as Mellor et al. (2021):

$$J_{i\alpha}(\mathbf{x}) = \partial_{\theta_\alpha} z_i^{(L)}(\mathbf{x}) \tag{4}$$

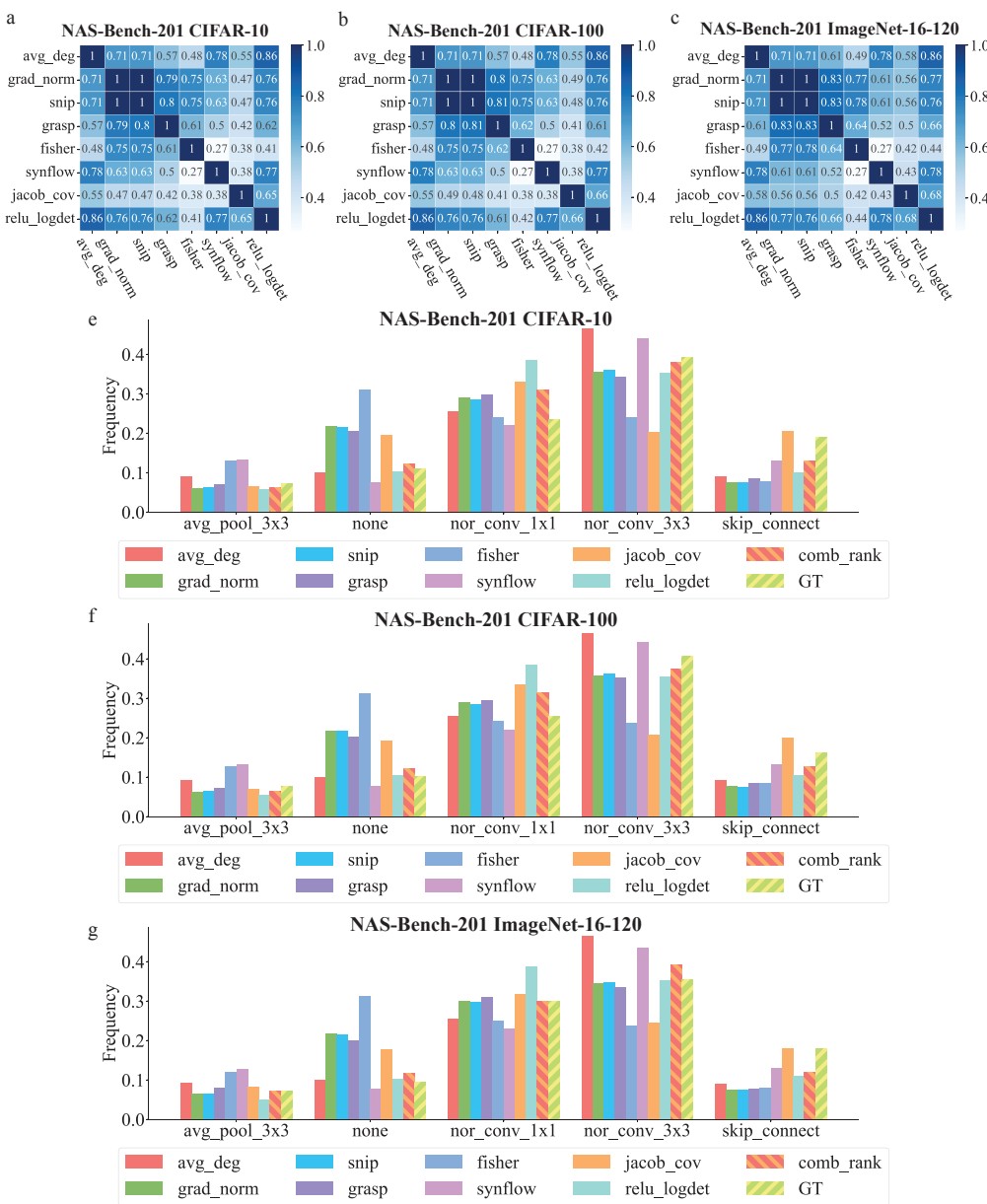

Figure 3: (a)-(c) Spearman's ranking correlation $\rho$ between pair of metrics and (e)-(f) the preference for different operations on NAS-Bench-201. GT is the operation distribution of top architectures ranked by test accuracy.

The `jacob_cov` metric takes gradient of model parameters into consideration, i.e., it focuses on the backward propagation process. The *NASGraph* framework, on the other hand, considers the forward propagation for each graph block. They are complementary to each other. Hence, Combining `avg_deg` with `jacob_cov` leads to a higher ranking correlation. `comb_rank` is rank(`avg_deg`) + rank(`jacob_cov`). The frequency distribution of `comb_rank` is very close to GT as shown in Figure 3 (e)-(g).

The `relu_logdet` metric exhibits the highest preference for convolution with small kernel size while GT indicates the best neural architectures prefer convolution with large kernel size. `relu_logdet` might have a problem when the search space $\mathcal{A}$ includes convolution with large

kernel size. ConvNeXt Liu et al. (2022) has shown the advantage of modern design using convolution with large kernel size. `fisher` has the highest preference for `none` operation, and it has the lowest correlation in the NAS-Bench-201.

## A.4 OUTPUT COMBINATION FROM PRECEDING GRAPH BLOCKS

When there are multiple graph blocks connected to the same graph block, there are two ways to combine the outputs from preceding graph blocks: *summation* and *concatenation*, as illustrated in the middle of Figure 1. In the case of summation, we do the forward propagation for each channel of all branches. In the case of concatenation, however, the outputs of the preceding graph blocks do not match the input dimension of the current graph block. We add virtual channels to the outputs of the preceding graph blocks such that the output dimension matches the input dimension, and hence we can do the conversion in the block-wise fashion. We want to emphasize that the summation or concatenation is determined by the original neural architecture. The graph block just combines components in the neural architecture such as Conv and ReLU. $w.l.o.g.$ we examine the case of concatenation and the case of summation of two preceding graph blocks connecting to the $l$-th graph block as shown in Figure 4. In the case of summation, two graph blocks connect to the $l$-th graph block. The output of each graph block has 4 channels. The input of $l$-th graph block has 4 channels. In the case of concatenation, two outputs from the preceding graph blocks have 2 channels while the input of $l$-th graph block has 4 channels.

**Summation** The summation requires that outputs of the preceding graph blocks have the same dimension as the input of the current graph block. We perform forward propagation for each channel of the outputs of the preceding graph blocks. Hence, there are 8 forward propagations in the case shown in Figure 4. Each forward propagation determines the scores between the "activated" channel and all the output channels of $l$-th graph block.

**Concatenation** The concatenation requires that outputs of the preceding graph blocks have the same dimension as the input of the current graph block except for the channel dimension, and the summation of the channel size of the outputs of the preceding graph blocks equal to the channel size of the input of the current graph block. Because we do the conversion in the block-wise fashion, there will be dimension mismatch if we do forward propagation using the dimension of the preceding outputs. So we add virtual channels to make sure two dimensions match. The number of added channels is $C^{(l)} - C^{(l-1)}$. In the case of Figure 4, two virtual channels are added for each input. We do not activate "virtual channels" in any forward propagation, so there are 4 forward propagations in total.

## A.5 PARALLEL COMPUTING FOR GRAPH BLOCKS

The direct way to compute the weight of a graph edge is to do forward propagation as many times as the number of input channels, and in this case, the batch size of the input is one. In other words, to compute the score $\omega_{i^{(l-1)}j^{(l)}}$, the input $\mathcal{M}_c^{(l)} \odot x^{(l)}$ has the batch dimension of one. To enable parallel computing of scores, we concatenate inputs for the same graph block on the batch dimension $[\mathcal{M}_1^{(l)} \odot x^{(l)}, \dots, \mathcal{M}_c^{(l)} \odot x^{(l)}]$ so that scores can be computed independently and in parallel. The batch size, under this condition, is equal to the input channel size. Using this approach, the batch size dimension has a different meaning compared to the standard definition in batch normalization. Moreover, the effective batch dimension is essentially one as we only use the same input $\mathbf{x}^{(l)}$ to determine scores. Therefore, during graph conversion, we remove batch normalization in the entire neural architecture. For example, Conv-BN-ReLU becomes Conv-ReLU.

## A.6 ALBATION STUDY

We use ablation study to analyze the effect of using surrogate models. There are two reducing factors: (1) number of channels, (2) number of search cells within one module. The random search result on NAS-Bench-201 is listed in Table 8 for the case $N = 100$ and $N = 200$. GT reports the highest test accuracy of neural architectures within the selected subset in the random search process. In both $N = 100$ and $N = 200$ cases, we do not find a significant variation of the performance of surrogate models except for the efficiency. As we decrease the number of cells or number of channels, there is a significant improvement in the efficiency. We use a grid routine to systematically study the effect

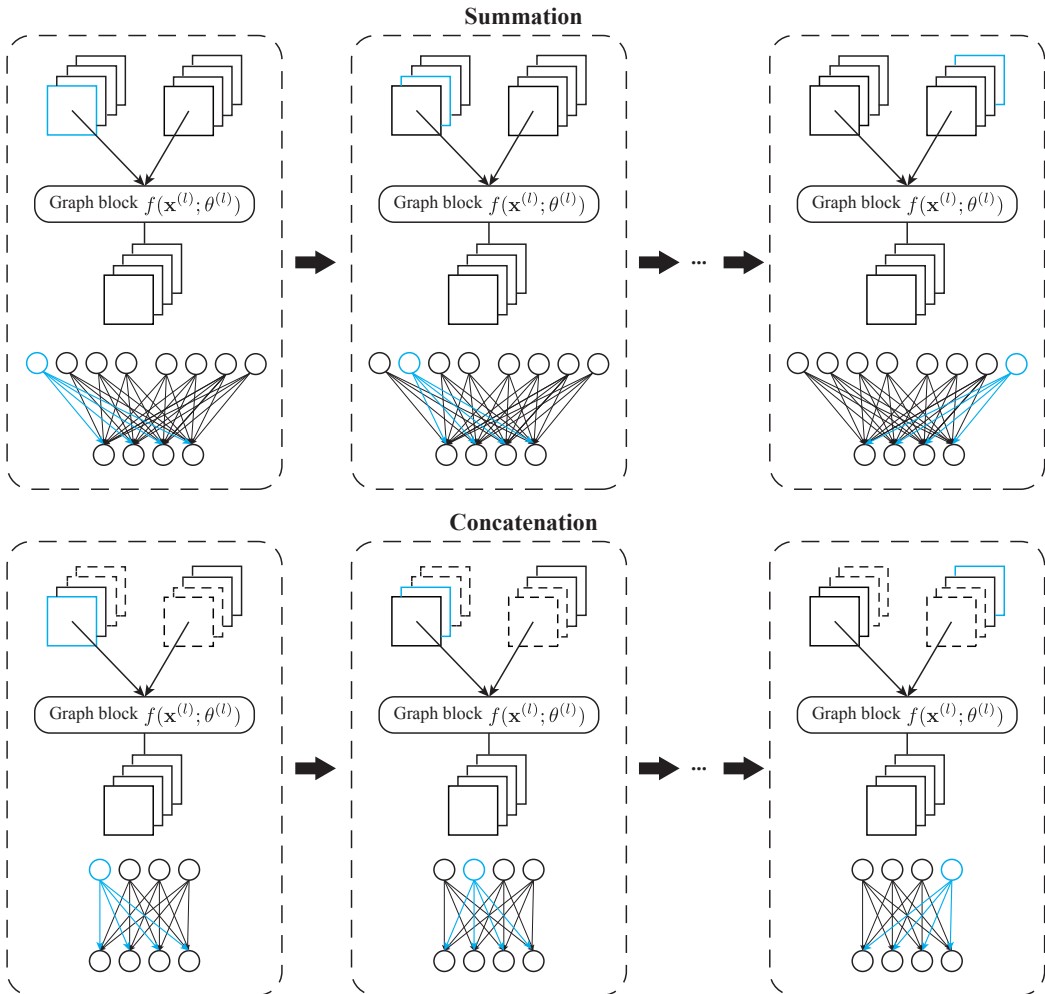

Figure 4: An illustration of converting one graph block of the neural architecture to subgraph in the case of summation and concatenation. Channels in dashed line mean virtually concatenated channels. These channels become zero matrix after applying a mask. Connection is established if the score between two graph nodes is non-zero.

of number of channels and number of cells on the performance of the surrogate model. Figure 5 shows the effect of varying number of channels and number of cells within the same module on NAS-Bench-201 using random search algorithm. The number of sampled architectures is 100 and each search process is repeated for 100 times.

**Number of channels** Along $x$ axis in Figure 5 shows the effect of varying the number of channels. We find as the number of channels increase, the performance of the surrogate models either does not change or improves. This is expected because decreasing the number of channels make the equivalent neural architecture has less model complexity (number of model parameters decrease) compared to the original one.

**Number of cells** Along $y$ axis in Figure 5 shows the effect of changing the number of cells within a module. We find there is no monotonous increase nor decrease in the performance as the number of cells varies. We believe it is related to the way we bridge neural architecture space and graph space. Because we convert neural architectures in the block-wise fashion and input is independent of other graph blocks, the conversion of one graph block is independent of other graph blocks. Besides, the cell structure is the same within the same module. Considering the cell structure that corresponds to a

subgraph, the module consisting of a linear stack of cell structures corresponds to a stack of identical subgraphs. Therefore, graph measures using surrogate models that decrease the number of cells is expected to not impose a significant effect on the graph measures, and hence rankings of the original neural network models.

Overall, we find that there is no remarkable difference in the accuracies using the random search algorithm. However, we can vastly boost the efficiency by the means of surrogate models. We choose the surrogate model *NASGraph*(16, 1, 3) to compute graph measures on different NAS benchamrks.

Table 8: Comparison of different surrogate models on NAS-Bench-201. `avg_deg` is used as the metric to score architectures. Reported results are averaged over 100 runs, and both mean values and standard deviations are recorded. GT records the highest accuracies of the randomly sampled architectures.

| Method | Time | CIFAR-10 | | CIFAR-100 | | ImageNet-16-120 | |
|---|---|---|---|---|---|---|---|
| | | validation | test | validation | test | validation | test |
| | | | | N = 100 | | | |
| *NASGraph*(1, 1, 3) | 7.78 sec. | $89.74 \pm 0.77$ | $89.53 \pm 0.75$ | $69.90 \pm 1.38$ | $70.01 \pm 1.43$ | $42.00 \pm 2.80$ | $40.73 \pm 4.14$ |
| *NASGraph*(4, 1, 3) | 19.23 sec. | $89.91 \pm 0.49$ | $89.70 \pm 0.52$ | $70.05 \pm 1.16$ | $70.22 \pm 1.18$ | $42.60 \pm 2.43$ | $43.00 \pm 2.42$ |
| *NASGraph*(8, 1, 3) | 62.61 sec. | $89.96 \pm 0.46$ | $89.74 \pm 0.49$ | $70.15 \pm 1.01$ | $70.28 \pm 1.05$ | $42.72 \pm 2.33$ | $43.15 \pm 2.30$ |
| *NASGraph*(16, 1, 3) | 106.18 sec. | $89.95 \pm 0.49$ | $89.73 \pm 0.52$ | $70.17 \pm 1.06$ | $70.29 \pm 1.10$ | $42.72 \pm 2.33$ | $43.15 \pm 2.29$ |
| *NASGraph*(1, 5, 3) | 25.63 sec. | $89.78 \pm 0.64$ | $89.58 \pm 0.65$ | $69.93 \pm 1.23$ | $70.06 \pm 1.30$ | $41.80 \pm 2.66$ | $42.22 \pm 2.63$ |
| *NASGraph*(4, 5, 3) | 89.68 sec. | $89.84 \pm 0.53$ | $89.62 \pm 0.55$ | $69.88 \pm 1.07$ | $70.04 \pm 1.13$ | $42.22 \pm 2.56$ | $42.66 \pm 2.56$ |
| *NASGraph*(8, 5, 3) | 258.03 sec. | $89.84 \pm 0.52$ | $89.62 \pm 0.55$ | $69.90 \pm 1.07$ | $70.07 \pm 1.14$ | $42.18 \pm 2.57$ | $42.65 \pm 2.57$ |
| GT | - | $90.98 \pm 0.36$ | $90.77 \pm 0.31$ | $71.48 \pm 0.86$ | $71.69 \pm 0.81$ | $45.45 \pm 0.67$ | $45.74 \pm 0.65$ |
| | | | | N = 200 | | | |
| *NASGraph*(1, 1, 3) | 15.98 sec. | $89.92 \pm 0.61$ | $89.69 \pm 0.62$ | $70.25 \pm 1.20$ | $70.42 \pm 1.21$ | $41.96 \pm 2.44$ | $42.48 \pm 2.39$ |
| *NASGraph*(4, 1, 3) | 61.60 sec. | $89.95 \pm 0.38$ | $89.73 \pm 0.42$ | $70.17 \pm 1.01$ | $70.42 \pm 0.99$ | $42.25 \pm 2.33$ | $42.73 \pm 2.32$ |
| *NASGraph*(8, 1, 3) | 186.21 sec. | $89.97 \pm 0.37$ | $89.74 \pm 0.42$ | $70.25 \pm 0.89$ | $70.47 \pm 0.87$ | $42.40 \pm 2.11$ | $42.87 \pm 2.13$ |
| *NASGraph*(16, 1, 3) | 217.21 sec. | $89.96 \pm 0.38$ | $89.73 \pm 0.43$ | $70.22 \pm 0.99$ | $70.45 \pm 0.98$ | $42.27 \pm 2.36$ | $42.76 \pm 2.36$ |
| *NASGraph*(1, 5, 3) | 50.33 sec. | $89.80 \pm 0.59$ | $89.61 \pm 0.59$ | $70.05 \pm 1.15$ | $70.18 \pm 1.23$ | $41.47 \pm 2.50$ | $41.97 \pm 2.51$ |
| *NASGraph*(4, 5, 3) | 196.08 sec. | $89.89 \pm 0.45$ | $89.68 \pm 0.49$ | $70.05 \pm 0.93$ | $70.21 \pm 1.05$ | $41.91 \pm 2.04$ | $42.45 \pm 2.05$ |
| *NASGraph*(8, 5, 3) | 413.47 sec. | $89.88 \pm 0.45$ | $89.66 \pm 0.49$ | $70.05 \pm 0.95$ | $70.23 \pm 1.07$ | $42.01 \pm 2.07$ | $42.53 \pm 2.09$ |
| GT | - | $91.14 \pm 0.25$ | $90.91 \pm 0.24$ | $71.84 \pm 0.76$ | $72.04 \pm 0.72$ | $45.72 \pm 0.54$ | $46.01 \pm 0.50$ |

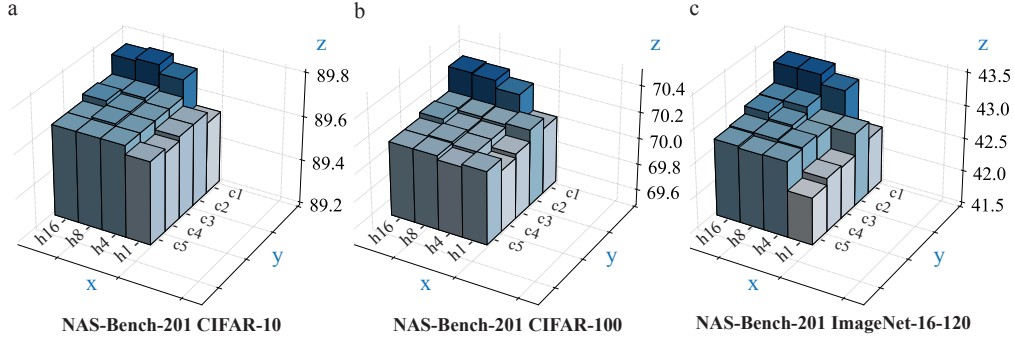

Figure 5: 3D-Bar chart for the reducing factors. The number of channels $h$ and number of cells $c$ within a module change in the ablation study. We examine 4 different number of channels along the $x$ axis: $h \in \{1, 4, 8, 10\}$ and 5 different number of cells along the $y$ axis: $c \in \{1, 2, 3, 4, 5\}$. The $z$ axis is the test accuracies of neural architectures.

## A.7 VARIATION IN RANKINGS USING DIFFERENT PARAMETER INITIALIZATION

Because the training-free NAS methods do a single forward/backward propagation on models with randomly initialized parameters, it is potentially subjected to different random initialization. To examine the variation in rankings due to different initialization of model parameters, we repeat the computation of metrics for 8 times. Each time, a different initialization is used, and the initialization follows normal distribution. We use the pair rank difference to indicate the variation in the rankings for a pair of two random processes. The pair rank difference is defined by:

$$\text{pair rank difference} = \sum_{k=0}^{n} |\text{rank}_i(a_k) - \text{rank}_j(a_k)| \tag{5}$$

where $n$ is the total number of neural architectures, on NAS-Bench-201 benchmark, $n = 15,625$. $\text{rank}_i(a_k)$ and $\text{rank}_j(a_k)$ are the ranking of $k$-th architecture in the $i$-th and $j$-th random initialization processes, respectively. Considering the computational overheads, we choose NASWOT, a NAS method requires training dataset, to compute the variation in rankings in comparison with our method. We use 8 random seeds and compute the pair rank difference. Figure 6 shows the pair rank difference on NAS-Bench-201 and different datasets. Because our method is data-agnostic, it has the same mean and standard deviation across different datasets. Our metric has a contiguously smaller variation in the rankings of neural architectures. We believe one of the reasons is that the NASWOT method, or other data-dependent NAS methods, has a random selection of minibatches and the random initialization of model parameters. Our method, on the other hand, is not subjected to the random selection of minibatches by using fixed inputs. When we compute ranking correlations between the performance of neural architectures and graph measures, we find there is negligible difference among 8 random initialization. So we believe that our method is not significantly subjected to the different random initialization.

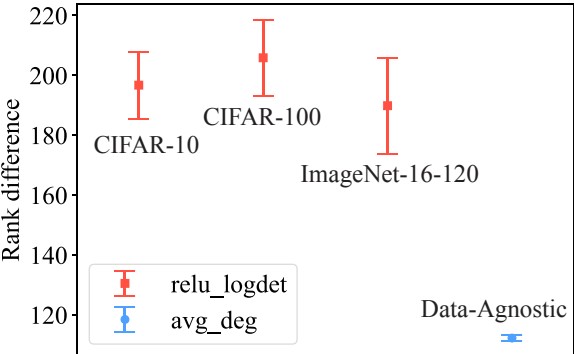

Figure 6: The variation of the architecture rankings using `relu_logdet` and `avg_deg` as the ranking metric. The mean value and standard deviation is calculated over 8 random seeds.

## A.8 EVALUATION OF DIFFERENT GRAPH MEASURES ON NAS BENCHMARKS

In addition to average degree, we also examine other graph measures in the training-free NAS. Those graph measures even exhibit a better performance than the average degree on some benchmarks.

**Graph measures.** After converting neural architectures to graphs $G(V, E)$ ($|V| = n$ and $|E| = m$) using *NASGraph*, we compute four graph measures as new metrics to rank neural architectures in NAS benchmarks, namely, average degree, density, resilience parameter, and wedge count. ① The **average degree** $\bar{k}$ calculates the average number of edges for one graph node. To compute the average degree for a DAG, we ignore the direction of the graph edges. We have $\bar{k} = \frac{1}{n} \sum_{i \in V} k_i$, where $k_i$ is the degree of node $i$. ② The **density** $d_G$ measures the ratio of the total number of edges to the maximum number of possible edges, $d_G = \frac{m}{n(n-1)}$. ③ The **resilience parameter** $\beta_{\text{eff}}$ of a DAG Gao et al. (2016) is defined by $\beta_{\text{eff}} = \frac{\mathbf{1}^T \mathbf{A} s^{\text{in}}}{\mathbf{1}^T \mathbf{A} \mathbf{1}} = \frac{\langle s^{\text{out}} s^{\text{in}} \rangle}{\langle s \rangle}$, where $\mathbf{1} = (1, \ldots, 1)^T$ is the all-ones vector, $\mathbf{s}^{\text{in}} = (s_1^{\text{in}}, \ldots, s_n^{\text{in}})$ is the vector of incoming degrees, and $\mathbf{A}$ is the adjacency matrix of the graph. ④ The **wedge count** $\mathcal{W}_G$ counts the number of wedges Gupta et al. (2014), and a wedge is defined as a two-hop path in an undirected graph. It is related to the triangle density of an undirected graph. To compute the wedge count of a DAG, we ignore the edge direction and use $\mathcal{W}_G = \sum_{i \in V} \frac{1}{2} k_i(k_i - 1)$. Table 9 summarizes these graph measures and their computation complexity.

Figure 2 shows the evaluation on average degree and density. The ranking correlations for all 4 graph measures (average degree $\bar{k}$, density $d_G$, resilience parameter $\beta_{\text{eff}}$ and wedge count $\mathcal{W}_G$) on NAS-Bench-101 and NAS-Bench-201 are shown in Figure 7. The surrogate model we use is *NASGraph*(16,

Table 9: Definition and computation complexity of the four graph measures used in *NASGraph*.

| | Average degree | Density | Resilience parameter Gao et al. (2016) | Wedge count Gupta et al. (2014) |
|---|---|---|---|---|
| Definition | $\bar{k} = \dfrac{1}{n}\sum_{i\in V} k_i$ | $d_G = \dfrac{m}{(n-1)}$ | $\beta_{\text{eff}} = \dfrac{\mathbf{1}^T\mathbf{A}\mathbf{s}^{\text{in}}}{\mathbf{1}^T\mathbf{A}\mathbf{1}}$ | $\mathcal{W}_G = \sum_{i\in V}\dbinom{k_i}{2}$ |
| Time complexity | $O(m+n)$ | $O(m+n)$ | $O(n^2+m)$ | $O(m+n)$ |

1, 3). On NAS-Bench-101, density is the best graph measure. On NAS-Bench-201, average degree is the best graph measure across different datasets. The ranking correlation for the same graph measure does not change significantly across three datasets on NAS-Bench-201, indicating the good generality of these graph measures. The average degree and density are correlated since they are all related to the number of graph edges within a unit (total number of graph nodes or maximum number of possible edges). On NAS-Bench-201, the difference in the ranking correlation between average degree and density is marginal. However, the difference becomes larger on NAS-Bench-101. The density essentially consider the graph $G(V, E)$ as directed graph while the average degree take it as undirected graph (since we remove directionality of the graph edge). As the NAS benchmark size increases, considering $G(V, E)$ as undirected graph might be inferior given the fact that information flow in the neural architecture is directional and acyclic (Note: we are not discussing recurrent neural networks in this paper).

Table 10: Comparison between the surrogate models *NASGraph*(16, 1, 3) and *NASGraph*(4, 5, 3) across different benchmarks and datasets.

| Methods | Metric | NAS-Bench-101 | | NAS-Bench-201 | | | | | |
|---|---|---|---|---|---|---|---|---|---|
| | | CIFAR-10 | | CIFAR-10 | | CIFAR-100 | | ImageNet-16-120 | |
| | | $\rho$ | $\tau$ | $\rho$ | $\tau$ | $\rho$ | $\tau$ | $\rho$ | $\tau$ |
| *NASGraph*(4, 5, 3)† | density | 0.50 | 0.35 | 0.75 | 0.56 | 0.77 | 0.57 | 0.75 | 0.55 |
| | avg_deg | 0.38 | 0.27 | 0.76 | 0.57 | 0.78 | 0.58 | 0.76 | 0.56 |
| | resilience | 0.34 | 0.23 | 0.74 | 0.54 | 0.75 | 0.55 | 0.73 | 0.53 |
| | wedge | 0.34 | 0.24 | 0.76 | 0.57 | 0.78 | 0.58 | 0.76 | 0.56 |
| *NASGraph*(16, 1, 3) | density | 0.45 | 0.31 | 0.77 | 0.57 | 0.78 | 0.59 | 0.75 | 0.56 |
| | avg_deg | 0.38 | 0.26 | 0.78 | 0.58 | 0.80 | 0.60 | 0.77 | 0.57 |
| | resilience | 0.31 | 0.21 | 0.68 | 0.49 | 0.69 | 0.50 | 0.67 | 0.48 |
| | wedge | 0.33 | 0.23 | 0.77 | 0.57 | 0.79 | 0.59 | 0.76 | 0.56 |
| Optimal single metric | | 0.50 | 0.35 | 0.78 | 0.58 | 0.80 | 0.60 | 0.77 | 0.57 |

† Number of cells within one module is 3 on NAS-Bench-101 while 5 on NAS-Bench-201. Because we only reduce the number channels, the surrogate model for NAS-Bench-101 is *NASGraph*(4, 3, 3) instead of *NASGraph*(4, 5, 3) on NAS-Bench-201.

Table 11: Ranking correlations $\rho$ between the validation accuracies and *NASGraph* metrics using different surrogate models. Because the structure of modules and cells is encoded in the arch string such as `64-41414-1_02_333` (we refer the reader to Duan et al. (2021) for details on the arch string), we only change the number of channels.

| Metric | Micro TransNAS-Bench-101 | | | | | | | |
|---|---|---|---|---|---|---|---|---|
| | class_object | | class_scene | | room_layout | | segment_semantic | |
| | $\rho$ | $\tau$ | $\rho$ | $\tau$ | $\rho$ | $\tau$ | $\rho$ | $\tau$ |
| | 4 Channels | | | | | | | |
| avg_deg | 0.56 | 0.39 | 0.71 | 0.51 | 0.38 | 0.25 | 0.67 | 0.48 |
| density | 0.54 | 0.37 | 0.68 | 0.49 | 0.37 | 0.24 | 0.60 | 0.43 |
| resilience | 0.62 | 0.44 | 0.75 | 0.55 | 0.47 | 0.31 | 0.33 | 0.23 |
| wedge | 0.60 | 0.42 | 0.74 | 0.54 | 0.43 | 0.28 | 0.68 | 0.49 |
| | 16 Channels | | | | | | | |
| avg_deg | 0.55 | 0.38 | 0.70 | 0.50 | 0.37 | 0.24 | 0.66 | 0.47 |
| density | 0.53 | 0.36 | 0.68 | 0.48 | 0.35 | 0.22 | 0.55 | 0.39 |
| resilience | 0.62 | 0.43 | 0.74 | 0.54 | 0.47 | 0.31 | 0.34 | 0.24 |
| wedge | 0.59 | 0.41 | 0.74 | 0.54 | 0.42 | 0.27 | 0.68 | 0.49 |

As indicated in Figure 5, decreasing number of cells or number of channels does not cause a huge effect on the rankings of graph measures. In addition to reduce the number of cells (the surrogate model *NASGraph*(16, 1, 3)), we also examine the effect of reducing the number of channels (the surrogate model *NASGraph*(4, 5, 3)). The comparison of the ranking correlation using different surrogate models is shown in the Table 10. These two surrogate models, as expected, have a similar ranking correlation across NAS-Bench-101 and NAS-Bench-201.

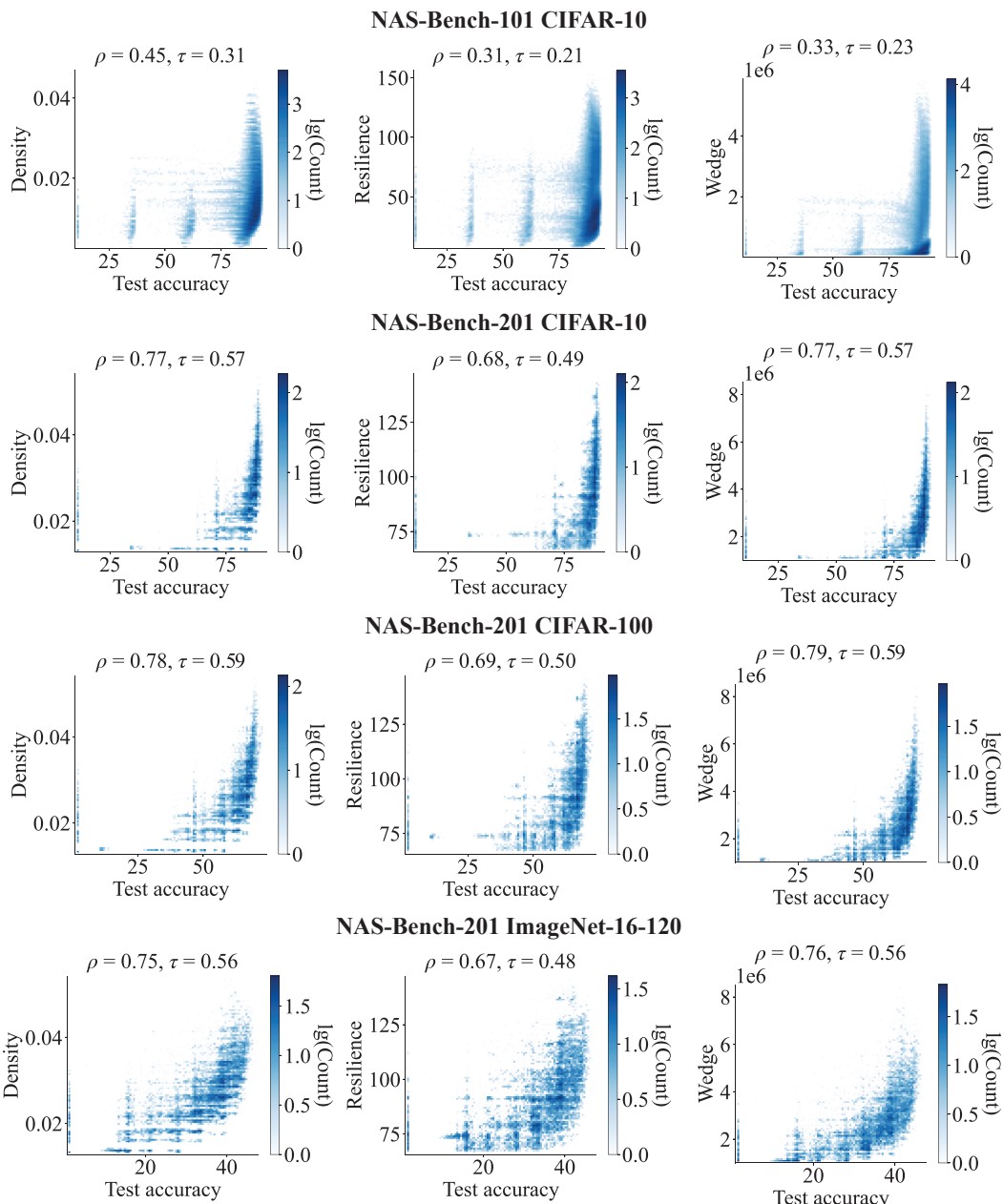

Figure 7: The ranking correlations between graph measures of the converted graphs and test accuracy of the corresponding neural architectures on NAS-Bench-201. 3 graph measures (*NAS-Graph* metrics) are examined: average degree (`avg_deg`), density (`density`), resilience parameter (`resilience`) and wedge count (`wedge`).

In addition to these two benchmarks, we also examine the performance of different surrogate model on TransNAS-Bench-101. Because the number of cells and modules are fixed in the arch string (the way to represent neural architecture on the benchmark), we only change the number of channels. Table 11 shows the evaluation of two surrogate models. There are only marginal difference between two graph measures across three tasks.

Overall, as indicated in the ablation study, decreasing number of channels and decreasing number of cells do not have a significant change in the rankings of graph measures. Based on graph theory, decreasing number of cells is preferred.

A.9 COMBINATION OF TRAINING-FREE NAS METRICS

We combine graph measure with training-free NAS metrics (rank(`avg_deg`) + rank(`jacob_cov`)) to boost the ranking correlation between metrics and performance of neural architectures. Figure 8 shows the relationship between combined ranks and test accuracies across CIFAR-10, CIFAR-100 and ImageNet-16-120 datasets. We note that `avg_deg` is data-agnostic while `jacob_cov` is data-dependent.

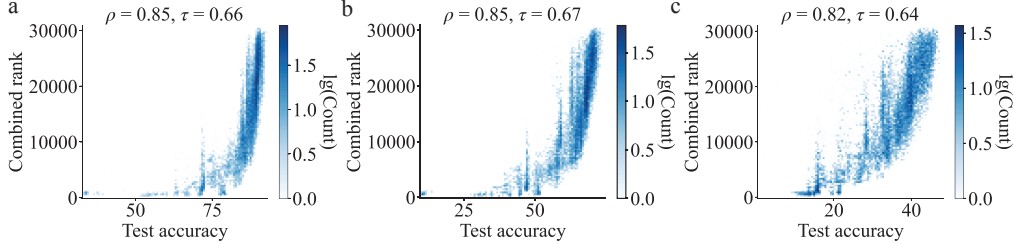

Figure 8: Combined rank (rank(`avg_deg`) + rank(`jacob_cov`)) of neural architectures of NAS-Bench-201 vs test accuracy. (a) CIFAR-10. (b) CIFAR-100. (c) ImageNet-16-120.

A.10 RANDOM SEARCH IN NAS

Algorithm 1 shows the process of random search using a single metric. A total number of $N$ neural architectures are randomly sampled from the same NAS benchmark such as NAS-Bench-101. Metrics are computed as the score by a single forward/backward propagation of neural architectures with randomly initialized parameters. In the *NASGraph* framework, we convert neural architectures to graphs and then compute graph measures such as average degree to rank neural architectures. The performance, e.g. test accuracy, of the neural architecture with the highest graph measure is extracted as the performance of the metric. The highest performance of the selected $N$ architectures are used as GT.

---

**Algorithm 1** Random Search Algorithm Using Single Metric

---

1: net_generator = RandomGenerator()
2: score_highest, net_best = None, 0
3: **for** $i = 1 : N$ **do**
4:     net = net_generator.pick_net()
5:     score = ComputeMetric(net)
6:     **if** score > score_highest **then**
7:         score_highest = score
8:         net_best = net
9:     **end if**
10: **end for**
11: acc_best = ExtractAccFromBenchmark(net_best)

---

A.11 CELL STRUCTURE

We visualize the best and the worst cell structures ranked by `avg_deg` on NAS-Bench-201 as shown in Figure 9. Edge with `none` operation is not shown for better visualization. The best cell structure found by our metric is same as `synflow`. The worst cell structures share a same feature: there is an isolated node of cell structure. The isolated node in NAS-Bench-201 means extracted features from preceding neural layers are disregarded, and the effective depth of neural architecture becomes shallower. Therefore, it is expected those architectures have a poor performance.

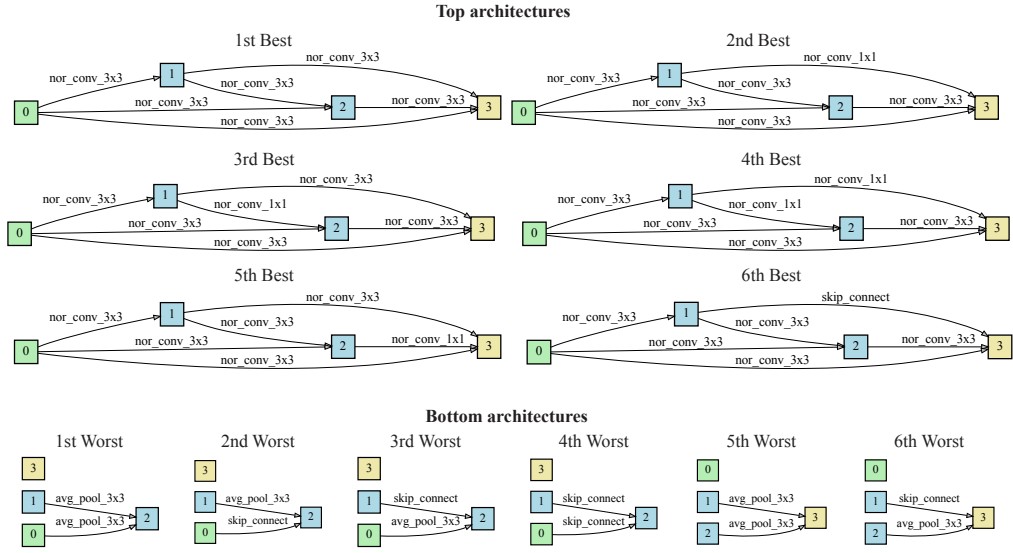

Figure 9: The top 6 best and top 6 worst cell structures ranked by average degree of *NASGraph* on NASBench-201. Architectures are ranked by the graph measure avg_deg.

## A.12 SPECIAL NASGRAPHS

Figure 10 shows the visualization of the converted graphs using the surrogate model *NASGraph*(4, 3, 5) corresponding to the best architecture and the worst architecture (ranked by the test accuracy) in NAS-Bench-101 and NAS-Bench-201. Instead of ranking architectures by the graph measures (as the ranking metric in Figure 2), we rank architectures by the test accuracy. As indicated by graph measures, the best graph is much denser compared to the worst graph.

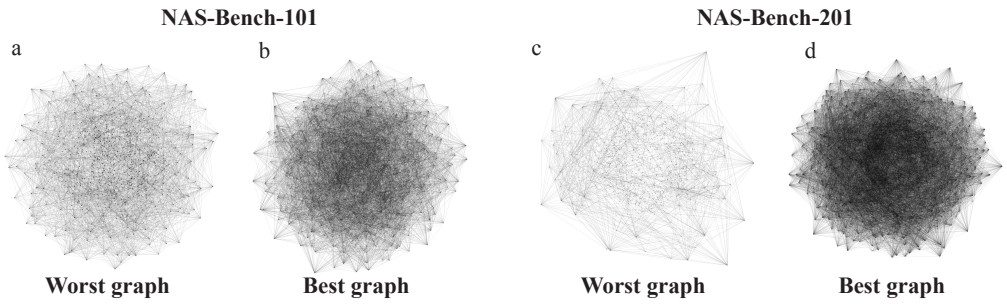

Figure 10: The converted graphs corresponding to the best architecture and the worst architecture on NAS-Bench-101 and NAS-Bench-201. Architectures are ranked by the test accuracy.

### A.13 COMPARISON TO TRAINING-BASED METHOD

We compare the total time of obtaining the performance of the optimal architecture with BONAS Shi et al. (2020), a training-based method. We use NAS-Bench-201 as our search space. The training time is estimated by the total number of evaluated architectures times the training time for one architecture. The accuracy is the highest test accuracy of the searched architectures. The result is shown in Figure 11. When the time limit for the searching and training is limited, our method (training-free) shows the better performance. If there is no time limit, the training-based method shows superior performance.

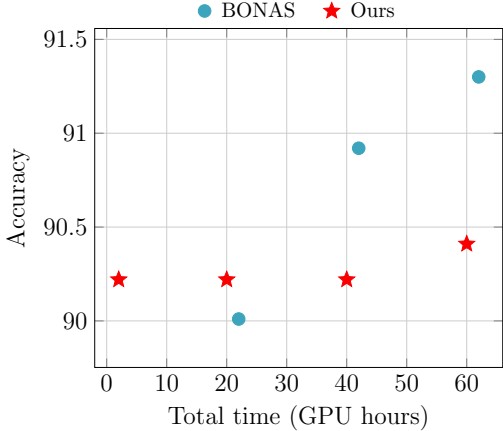

Figure 11: The comparison of total time to BONAS Shi et al. (2020).