# OpenReview forum: "Graph is All You Need? Lightweight Data-agnostic Neural Architecture Search without Training"
_ICLR.cc/2024/Conference — Submitted to ICLR 2024_

### Official Review · Reviewer_UMqx · 2023-10-31

**Soundness:** 3 good
**Presentation:** 3 good
**Contribution:** 2 fair
**Rating:** 5
**Confidence:** 3

**Summary:**

- The authors propose a novel, cheap, training-free, and data-agnostic neural architecture search (NAS) method.
- The proposed method converts an architecture to a graph and utilizes extracted graph measures such as average degree as a proxy for the performance of the architecture.
- The empirical results show that the proposed method can find architectures with better performance in computational costs compared to the baseline methods.
- Surprisingly, the proposed method can find the best architecture among 200 randomly sampled architectures from NAS-Bench 201 dataset in 217 CPU seconds without any GPU computations.

**Strengths:**

- The proposed method is simple and easy to understand.
- The proposed conversion method seems novel.
- The authors suggest techniques for further speeding up the proposed method by using surrogate models.
- The empirical results show the superiority of the proposed method.
- The paper contains the comparison among various graph measures in its appendix.

**Weaknesses:**

- I think the paper needs more justification about the importance of the training-free data-agnostic NAS.
- The experimental results are evaluated on only simple benchmarks. To show the practicality of the proposed method, I think a it would be more helpful if you provide a comparison with other non-training-free NAS methods such as DARTS-PT on a larger search space such as DARTS space.

**Questions:**

- I am curious about the correlation between model inference time and proxy metrics. Since the proposed "average degree" measures the connectivity between channels, a higher average degree may lead to a slower inference speed.
- The ultimate goal of NAS is to find the optimal architecture in any way possible. Even if the proposed method has a high correlation with model performance, it is uncertain whether it will be significantly helpful in quickly finding the best architecture. This is because one must train selected architectures to evaluate the actual performance and verify whether it is a good architecture, and it seems that most of the time will be spent in this verification process during the NAS procedure. Can you kindly and elaborately explain why training-free NAS is an important issue, along with practical scenarios?
- Of course, unlike existing non-training-free NAS methods, since the proposed method has near-zero search cost, most of the time can be spent on validation. Nonetheless, I am curious whether it can find a better architecture than baselines such as DARTS-PT within the same computational budget in the DARTS space.

If my concerns are addressed, I promise to raise the score.

**Details Of Ethics Concerns:**

There aren't any ethical concerns related to this paper.

---

> ### Author Response · Authors · 2023-11-17
>
> We thank the reviewer for the comments. Below is the one-to-one response:
>
> - Weakness 1st point
>
> Neural architecture search finds candidate architectures, trains them to obtain their accuracy and uses the accuracy as feedback to modify its search strategy. NAS is computationally expensive and slow and training the architecture consumes the majority of the time. Training-free NAS techniques developed proxy scores in lieu of accuracy. This avoids the need to train the architecture altogether and the training-free metric is computed on a randomly initialized model. These metrics are relatively fast to compute. For instance, our average degree takes only 1 CPU second per architecture.
> The candidates are now ranked using these proxy scores instead. Thus, training-free NAS accelerates the search drastically.
>
> To save further time, a lot of these training-free NAS algorithms compute the proxy scores only on a few mini-batches of the data. This score might not generalize well across the entire dataset and has a high variance based on the mini-batches passed. To address all the above mentioned issues, we developed a data-agnostic training-free NAS method.
>
> - Weakness 2nd point
>
> In NAS, the search algorithm searches for a candidate architecture and uses its validation accuracy to determine the candidate’s efficacy. All the training-free metrics  are used in lieu of accuracy. Instead of ranking the candidate models based on the accuracy and selecting the best model found so far, one could use the proxy scores. These training-free metrics can be combined with any existing NAS algorithm. Thus it is essential that all these training-free metrics have a very high correlation between the ranking of the candidates by their score and the ranking by the validation accuracy. We already reported the correlation metrics on more complex search spaces such as ENAS, PNAS, DARTS etc.  in Table 4 of our paper.
>
> NDS-DARTS is a NAS Benchmark developed for DARTS search space. The accuracy of the best architecture found by our metric is 93.23% . NASWOT got 90.6% on the same search space. On NAS-Bench-201 search space, DARTS-PT reported a test accuracy of 88.11% (table 4 of the paper [2]). Our method using average degree yielded an architecture with 90.2% test accuracy.
>
> - Question 1st point
>
> Following the reviewer's comments, we record the inference time for architectures of 10 highest average degrees, 10 lowest average degrees, 10 highest test accuracies and 10 lowest test accuracies in NAS-Bench-201. In this case, the reviewer's intuition is right that the architectures correspond to a higher average degree tend to have a longer inference time. The inference time is recorded in the NAS-Bench-201 benchmark.
>
> |                                       |   High average degree | Low average degree | High test accuracy | Low test accuracy |
> | -------------------------- | ------------------------- | ----------------------- | ---------------------- | --------------------- |
> | Average inference time |            23.10 ms          |             10.44 ms       |            20.29 ms     |           13.74 ms      |
>
> - Question 2nd point
>
> In NAS, the search algorithm searches for a candidate architecture and uses its validation accuracy to determine the candidate’s efficacy. The goal of the algorithm is to find the architecture with the best test accuracy. Each candidate architecture is trained and then its validation accuracy is used as the feedback for the search algorithm. NAS is very computationally expensive, requiring every candidate architecture to be trained. To alleviate this problem, training-free NAS was introduced [3, 4, 5].
>
> The training-free metrics are used in lieu of accuracy. The training-free metric is computed on a randomly initialized model and is generally fast to compute. Instead of ranking the candidate models based on the validation accuracy and selecting the best model found so far, the candidate models are now ranked using this training-free metric.
>
> In particular, our average degree metric just randomly initializes the candidate architecture and computes the average degree of the model. This is extremely fast and takes only a ~1 CPU second for each model.
>
> - Question 3rd point
>
> NDS-DARTS benchmark [1]  is created by randomly sampling architectures from the DARTS search space. The best performing architecture in this search space has the test accuracy of 95.06 on CIFAR10. When we searched on the search space using the average degree as the metric, the test accuracy of the best model we found was 93.23%. Thus, our method generalizes to larger search spaces as well.
>
> DARTS-PT also performed architecture search on the NAS-Bench-201 benchmark and reported a test accuracy of 88.11% (Table 4 of their paper [2]). NAS using average degree on NAS-Bench-201 yielded an architecture with 90.2% test accuracy.

---

> > ### Comment · Reviewer_UMqx · 2023-11-20
> >
> > Thank you for addressing my inqueries.
> >
> > I think there might be a slight misinterpretation regarding the second point of my question.
> >
> > I'm fine with the search cost being 1 second. However, even if a promising set of candidates is quickly identified, to determine if these candidates are truly effective, end-to-end actual training is necessary to measure the test accuracy, which will consume most of the time and cost.
> >
> > Consider a hypothetical scenario:
> >
> > ======================================
> >
> > We aim to identify an architecture with a test accuracy higher than $c$. Suppose evaluating a single architecture requires 4 GPU hours. Assume that there are two NAS methods having the following properties.
> >
> > - Method A requires 1 second to rank whole candidates but the first architecture that has a test accuracy higher than $c$ is ranked 50th.
> > - Method B requires 100 GPU hours for searching but can find test accuracy higher than $c$ at once.
> >
> > =======================================
> >
> > In this context, Method A, despite its rapid ranking ability, results in an overall NAS time of 1 second + (4 GPU hours × 50), which exceeds Method B's total of 100 GPU hours + (4 GPU hours × 1).
> >
> > Can you provide experimental evidence or scenarios to validate the effectiveness of the zero-shot approach compared to non-zero-shot NAS methods (such as [1,2,3,4]) in the end-to-end NAS process? It can be illustrated through the plot comparing 'the best test accuracy versus total runtime', where 'total runtime' encompasses both the search time and test accuracy evaluation time."
> >
> > I am aware of the limited time remaining in the review period. However, for the camera-ready version, I recommend elaborating on why the proposed method is beneficial in the overall end-to-end process.
> >
> > [1] Bridging the Gap between Sample-based and One-shot Neural Architecture Search with BONAS, Shi et al., NeurIPS 2020.
> >
> > [2] Rethinking Architecture Selection in Differentiable NAS, Wang et al., ICLR 2021.
> >
> > [3] $\beta$-DARTS: Beta-Decay Regularization for Differentiable Architecture Search, Ye et al., CVPR 2022.
> >
> > [4] Shapley-NAS: Discovering Operation Contribution for Neural Architecture Search, Xiao et al., ICLR 2022.

---

> > > ### Author Response · Authors · 2023-11-21
> > >
> > > We thank the reviewer for the suggestion of this insightful experiment. We follow the reviewer’s comment to conduct an experiment on NAS-Bench-201 to report the best test accuracy found given total GPU hours. For a fair comparison, we only compare our method with BONAS [1], because other listed training-based NAS methods tend to find new architectures that are not included in NAS-Bench-201.
> > >
> > >
> > > We include the figure in the updated supplementary materials (Figure 11), we report the best test accuracy with respect to the total time (searching + training time) of our method versus BONAS [1]. The training time is evaluated by training one architecture using single GPU on CIFAR-10 dataset. The reviewer’s intuition is correct that if given plenty of compute resources and time, eventually training-based NAS will outperform training-free NAS. However, given limited compute time, training-free NAS and our method have the clear advantage.
> > >
> > > [1] Bridging the Gap between Sample-based and One-shot Neural Architecture Search with BONAS, Shi et al., NeurIPS 2020.

---

### Official Review · Reviewer_EKQj · 2023-11-03

**Soundness:** 2 fair
**Presentation:** 2 fair
**Contribution:** 3 good
**Rating:** 6
**Confidence:** 4

**Summary:**

The paper proposes NASGraph, a training-free Neural Architecture Search (NAS) method that relies on a graph-based interpretation of neural architectures. NASGrPh first converts neural architectures to computational graphs and then utilizes properties of the generated graphs as proxy of validation accuracies of corresponding architectures. NASGraph utilizes graph blocks to modularize neural architectures and determine the connectivity of graph nodes. The proposed conversion method allows NASGraph to construct a computational graph that reflects the forward propagation process of a neural architecture. The effectiveness of NASGraph is verified across various NAS benchmarks.

**Strengths:**

- While this is not the first work to perform architecture search with graph-based representations of neural architectures, I believe the proposed approach is far more thorough in incorporating the actual computations that occur within neural architectures. NASGraph goes above and beyond simply converting neural architectures into DAGs by considering how the inputs are being processed and mapped to outputs during the forward propagation process.
- The final proxy metric to the validation accuracy of neural architectures (the average degree of the graph measure) makes sense and is theoretically-grounded.
- The experiments that span various NAS benchmarks are extensive and comprehensive.

**Weaknesses:**

- It appears that NASGraph assumes the inputs to the neural architecture and subsequent graph blocks is non-negative. Does the analysis hold even with non-negative inputs? Many of modern neural architectures utilize activation functions that could yield negative inputs/outputs (e.g., gelu activation). Can NASGraph generalize beyond relu-based architectures?
- In a similar vein, can the proposed method generalize to non-conv-based architecture spaces? Such as ViTs and MLPMixers?
- The authors define the final score of neural architectures (the average degree of the graph measure) in Section 4.2 under the performance evaluation section. I think it would be more appropriate to move this definition to somewhere towards the end of Section 3 because by the end of Section 3, the reader is left hanging without the knowledge of how NASGraph actually ranks neural architectures.
- In Section 4.2, the authors mention that they explore other graph measures as well. Any idea why the average degree works best out of the compared graph measures? Also, intuitive explanation to what each one of these graph measures actually indicates/implies would be helpful.

**Questions:**

Please refer to the Weaknesses section.

---

> ### Author Response · Authors · 2023-11-17
>
> We thank the reviewer for the comments. Below is the one-to-one response:
>
> - Weakness 1st point
>
> When there is non-negative input to the graph block involved, we need a threshold to create non-negative edge weights in our graph. When the input is larger than the threshold, we consider it as active. Otherwise, we consider it as inactive. Therefore, our method can be applied to GeLU activation as well.
>
> - Weakness 2nd point
>
> Given that we convert a neural network to a graph and our proposed framework only considers the input/output tensor, in principle, our method is applicable to most search spaces. However, owing to the lack of publicly available NAS benchmarks for those search spaces, we cannot evaluate it on those search spaces right now.
>
> - Weakness 3rd point
>
> We appreciate this valuable suggestion. We will modify section 4.2 accordingly: the average degree is now introduced at the bottom of Section 3.
>
> - Weakness 4th point
>
> Intuitively, each graph edge indicates if the previous graph block (defined in the context of the NASGraph) contributes to the current graph block. An extreme case is that the graph block is Zero operation (used in NAS-Bench-201). In this case, the previous graph block does not contribute to the current graph block. There will be no graph edge connecting these two graph blocks. The performance of such an architecture is expected to be inferior. In short, the more the number of edges (higher average degree), the better the performance of the corresponding architecture would be. To further illustrate this, in Section A.12 of our paper, we plotted the average degrees of the architectures with the highest and the lowest accuracy in the NASBench 201 benchmark. As expected, the architecture with the highest accuracy has a very high average degree compared to the other one.
>
> In addition to the average degree, we explore other graph measures such as wedge, resilience, and density. We refer the reviewer to Section A.8 in the appendix.

---

### Official Review · Reviewer_bWfm · 2023-11-08

**Soundness:** 2 fair
**Presentation:** 3 good
**Contribution:** 3 good
**Rating:** 3
**Confidence:** 4

**Summary:**

Neural Architecture Search (NAS) is inherently as expensive task. Zero cost proxies aim at bypassing this compute by using cheap to evaluate statistics on architectures to predict the most optimal architectures. This paper proposes a novel graph-based zero cost proxy based on properties of graph representations of architectures.  The method is evaluated on multiple NAS-Bench-Suite Zero tasks yielding competitive performance.  The method overcomes several issues with current zero cost proxies e.g. data dependency, GPU requirement, and operation preference.

**Strengths:**

- The paper in general is well written and the results/evaluation well presented
- Results on the NAS-Bench-Suite-Zero tasks are quite competitive. Evaluation of complementary nature of the proxy, from table 5 is interesting.
- The method is fairly novel and interesting

**Weaknesses:**

- I went through the example in Figure 1 and the caption and the example itself is still unclear to me. Could the authors please elaborate on this?
- I find the evaluation of the method quite weak especially since the authors do not compare against the MeCo proxy https://openreview.net/pdf?id=KFm2lZiI7n. On an initial glance it seems that MeCo outperforms the proposed proxy on most of the benchmarks. The code for MeCo is publicly accessible and I encourage the authors to compare their work with MeCo in terms of the correlation metric and search time.
- I currently find the search spaces to be very limited to cell-based spaces or benchmarks. Since recently there have been efforts to scale Neural Architecture Search to weight-entangled spaces like AutoFormer [1], OFA [2] and HAT[3], it would be great to evaluate the method on these spaces. Note though these spaces don't have a tabular benchmark for evaluation, they do provide a pre-trained surrogate model for fast evaluation.
- Since there has been growing interest in transformer spaces, is the proxy search space agnostic and directly applicable to transformer spaces? This was unclear to me from the paper.

[1] Chen, M., Peng, H., Fu, J. and Ling, H., 2021. Autoformer: Searching transformers for visual recognition. In Proceedings of the IEEE/CVF international conference on computer vision (pp. 12270-12280).

[2] Cai, H., Gan, C., Wang, T., Zhang, Z. and Han, S., 2019. Once-for-all: Train one network and specialize it for efficient deployment. arXiv preprint arXiv:1908.09791.

[3] Wang, H., Wu, Z., Liu, Z., Cai, H., Zhu, L., Gan, C. and Han, S., 2020. Hat: Hardware-aware transformers for efficient natural language processing. arXiv preprint arXiv:2005.14187

**Questions:**

- Check weaknesses: Could the authors compare against MeCo (https://openreview.net/pdf?id=KFm2lZiI7n) and could the authors evaluated on weight entangled macro-architecture spaces like AutoFormer[1], OFA[2] and HAT[3]?
- How does the cuda memory consumption and search time of the method compare against other proxies?
- Reproducibility is still am important challenge in the NAS literature . For reproducibility, could the authors make the code publicly available?


[1] Chen, M., Peng, H., Fu, J. and Ling, H., 2021. Autoformer: Searching transformers for visual recognition. In Proceedings of the IEEE/CVF international conference on computer vision (pp. 12270-12280).

[2] Cai, H., Gan, C., Wang, T., Zhang, Z. and Han, S., 2019. Once-for-all: Train one network and specialize it for efficient deployment. arXiv preprint arXiv:1908.09791.

[3] Wang, H., Wu, Z., Liu, Z., Cai, H., Zhu, L., Gan, C. and Han, S., 2020. Hat: Hardware-aware transformers for efficient natural language processing. arXiv preprint arXiv:2005.14187

---

> ### Author Response · Authors · 2023-11-17
>
> We thank the reviewer for the comments. Below is the one-to-one response:
>
> - Weakness 1st point:
>
> As detailed in Sec. 3, we group neural layers (or operations such as conv, bn and relu) as graph blocks as shown in Figure 1 (a). Then, we do one single forward propagation to determine if the previous graph block contributes to the current graph block in a channel-wise manner. If there is a contribution, we build a graph edge. Otherwise, there is no graph edge. Each channel of the input tensor (or the output tensor) is aggregated to a graph node. This way, we are able to build a new type of graph given an architecture. The connectivity between graph nodes is determined by the forward propagation, i.e., outputs of graph blocks. We do a neural architecture search by comparing graph measures.
>
> - Weakness 2nd point
>
> We appreciate the suggestion of making comparisons to MeCo, a recent publication just accepted to NeurIPS 2023 and to be presented in December 2023. However, we want to bring to the reviewer’s attention that this is an unfair ask, because our ICLR submission was made back in Sep. 2023. To our best knowledge, the earliest public publication record of MeCo is Sep. 21st 2023, on NeurIPS’s Openreview system (we did not find any arxiv version). This situation makes it impossible to compare to MeCo at the time of our submission, not to mention that MeCo should be considered as a concurrent work. Despite the impossibilities, we follow the reviewer’s suggestion to include the comparison in the rebuttal and the updated manuscript. But we hope our review rating will not be negatively and unfairly judged by not having comparisons to a concurrent work that is only publicly available around the ICLR submission deadline.
>
> We compared our method with MeCo on  NAS-Bench-101, NAS-Bench-201, Trans-NASBench 101, NDS-DARTS and NDS-ENAS. While it is true that MeCo has competitive performance on NAS-Bench-101 and NAS-Bench-201 as reported in the MeCO paper, we observed that MeCo does not generalize well on other benchmarks. As stated in [1], [2], there is no one proxy that performs the best across all the search spaces. We compared MeCo on the Trans-NASBench 101 search space and evaluated it on the NDS-DARTS and NDS-ENAS search spaces, as those search spaces were already reported in our paper. We used one A40 NVIDIA GPU and one AMD EPYC 7232P CPU to run Meco. Our method runs only on a CPU and does not require a GPU.
>
> Trans-NASBench-101 results:
>
> |                                   | surface_norm | class_scene | room_layout  | Average Running Time  |
> |-------------------------|-----------------|---------------|-----------------| --------------------------- |
> | MeCo                        |      0.65           |        0.62       |       -0.25        |     6.88 GPU hours        |
> | MeCo opt                  |      0.67           |        0.64       |        0.26        |     6.90 GPU hours        |
> | Avg degree (ours)     |      0.66           |        0.70       |        0.37        |     2.93 CPU hours        |
> | Wedge (ours)            |      0.72           |        0.74       |        0.42        |     3.04 CPU hours        |
>
> It is important to note that while most metrics perform well on NASBench 101 and 201, their performance drops on Trans-NasBench-101.  We used the publicly available Meco code to evaluate it on NDS search space and the correlation is given in the table below.
>
> |                                |     NDS-DARTS*       |   NDS-ENAS*   | Average Running Time |
> | --------------------- | ----------------------- | ----------------- | -------------------------- |
> |  MeCo                    |            0.31              |         0.19          |        36.50 GPU hours   |
> | Avg degree (ours)  |            0.62              |         0.57           |         3.34 CPU hours    |
>
> *There is no performance reported on NDS search space, so we ran experiments using the publicly available code and computed the correlation. We did encounter some bugs while trying to run Meco’s code, but we fixed it to the best of our knowledge.
>
> [1] https://iclr-blog-track.github.io/2022/03/25/zero-cost-proxies/
> [2] NAS-Bench-Suite-Zero: Accelerating Research on Zero Cost Proxies, Krishnakumar et al.
>
> - Weakness 3rd point
>
> As the reviewer rightly pointed out, there are no publicly available benchmarks for AutoFormer, OFA and HAT. While OFA does have a surrogate model, the error introduced by the surrogate model would definitely hamper us from determining the true correlation. AutoFormer and HAT do not even have surrogate models to predict the accuracies given an architecture. We do not have the computational power to evaluate 1000 architectures in these search spaces to compute the correlation.

---

> > ### Author Response · Authors · 2023-11-17
> >
> > Below is the continuation of the previous comment:
> >
> > - Weakness 4th point
> >
> > In principle, our method is applicable to any neural architecture as it only considers the input/output tensors. The connectivity between graph nodes is determined by the forward propagation. Thus it can be run on the transformer search space too but we don’t have the computational power to evaluate 1000 architectures on Imagenet to compute the correlation. We understand the great interest in extending our method to transformer spaces. This is beyond the scope of this paper and is considered as one of the promising future works based on our study.
> >
> > - Question 1st point
> >
> > Same as the Weakness 2nd point, please refer to it.
> >
> > - Question 2nd point
> >
> > In the manuscript, we explicitly claim that we only need CPU for the computation. So there is no cuda memory consumption. In table 6, we report the running time needed for the random search.
> >
> > - Question 3rd point
> >
> > The code has already been shared publicly. The link for the git repo is listed in the supplementary material.

---

### Author Response · Authors · 2023-11-22
**A Gentle Reminder on Reviewers' Feedback**

Dear All Reviewers,

First of all, we would like to thank you all again for your valuable time and efforts spent reviewing our paper and helping us improve it. We have also revised our paper to reflect our responses to your questions.

As the discussion period is closing, we sincerely look forward to your feedback.

It would be very much appreciated if you could once again help review our responses and let us know if these address your concerns. We are eager to answer any further questions you might have. We strive to improve the paper consistently, and it is our pleasure to have your feedback!

Yours Sincerely,

Paper 6348 Authors

---

### Meta-Review · Area_Chair_tmGa · 2023-12-04

**Metareview:**

The scores for this paper were 3, 5 and 6. The reviewer giving a borderline accept did not engage in the private discussion, but the most negative reviewer posted a reply to the authors' rebuttal, which I'm replicating here to make it visible to the authors:

"I thank the authors for the detailed responses to each of my questions. I appreciate all the additional evaluations and comparison with MeCo. Few of my concerns which still remain after the rebuttal are as follows :

- The authors mention " there here are as no publicly available benchmarks for AutoFormer, OFA and HAT". While I agree that there are no tabular benchmarks for these spaces the pre-trained surrogates or the pre-trained supernets for all of these are publicly available. Hence search on these spaces is very cheap and doesn't need a lot of compute power.

- I think evaluation on these spaces (Transformers, MobileNet-V3) is very important to demonstrate the usefulness of zero-cost proxies to more realistic search spaces and applications, where one-shot NAS methods have traditionally been applied.

Since some of my concerns are not completely addressed I keep my score and encourage the authors to include comparisons on these spaces to the final version of their paper."

Overall, the reasons for rejection prevail. I recommend the authors to address the reviewers' concerns and resubmit to another venue.

**Justification For Why Not Higher Score:**

No reviewer is arguing for the paper.

**Justification For Why Not Lower Score:**

N/A

---

### Decision · Program_Chairs · 2024-01-16

Reject